# Assessing Local Climate Change by Spatiotemporal Seasonal LST and Six Land Indices, and Their Interrelationships with SUHI and Hot–Spot Dynamics: A Case Study of Prayagraj City, India (1987–2018)

**Md. Omar Sarif** [1,*] , **Rajan Dev Gupta** [2] **and Yuji Murayama** [3]

1 Department of Geography, Lovely Professional University, Phagwara 144411, India
2 Civil Engineering Department, Motilal Nehru National Institute of Technology Allahabad, Prayagraj 211004, India
3 Faculty of Life and Environmental Sciences, University of Tsukuba, 1-1-1 Tennodai, Tsukuba City 305-8572, Ibaraki, Japan
* Correspondence: omar.28426@lpu.co.in or mdomarsarif@gmail.com

**Abstract:** LST has been fluctuating more quickly, resulting in the degradation of the climate and human life on a local–global scale. The main aim of this study is to examine SUHI formation and hotspot identification over Prayagraj city of India using seasonal Landsat imageries of 1987–2018. The interrelationship between six land indices (NDBI, EBBI, NDMI, NDVI, NDWI, and SAVI) and LST (using a mono-window algorithm) was investigated by analyzing correlation coefficients and directional profiling. NDVI dynamics showed that the forested area observed lower LST by 2.25–4.8 °C than the rest of the city landscape. NDBI dynamics showed that the built-up area kept higher LST by 1.8–3.9 °C than the rest of the city landscape (except sand/bare soils). SUHI was intensified in the city center to rural/suburban sites by 0.398–4.016 °C in summer and 0.45–2.24 °C in winter. Getis–Ord $G_i$* statistics indicated a remarkable loss of areal coverage of very cold, cold, and cool classes in summer and winter. MODIS night-time LST data showed strong SUHI formation at night in summer and winter. This study is expected to assist in unfolding the composition of the landscape for mitigating thermal anomalies and restoring environmental viability.

**Keywords:** LST; mono-window algorithm; land indices; correlation coefficients; directional profiling; SUHI; hotspots (Getis–Ord $G_i$* statistics); MODIS night-time LST; Prayagraj city

## 1. Introduction

Globally, 55% of the total populace resided in urban areas in 2018, and prediction statistics show that if this trend continues, then the urban population will account for 68% of the total in 2050 [1]. Recently, the IPCC revealed in its report that the global mean surface temperature (GMST) increased by 1.53 °C and the global mean air temperature (GMAT) [both land and ocean] increased by 0.87 °C during the preindustrial period (1850–1900) and recent postindustrial period (2006–2015) [2]. The rise in land surface temperature (LST) has severe environmental consequences because unplanned urbanization will deteriorate climate equilibrium and hamper human life and health from microscale to macroscale [3,4]. In its AR6 2021 report, IPCC revealed that we are on the way to reaching 1.5 °C more global warming in the next 20 years. The unprecedented changes in the recent past have been highly challenging and alarming, leading to uncertain precipitation, increased glacier melting, mean sea level rising, floods, droughts, damage to agricultural land, and food shortage affecting every region of the globe [5].

The presence of Earth surface objects, such as asphalt, stones, pebbles, and sand, over the city landscape has diverse electromagnetic behavior in terms of evaporation, absorption, and radiation. Longwave radiation, as well as prevailing winds, assimilates into massive

heat discharge from the Earth's surface [6]. These surface objects weaken evapotranspiration and accelerate sensitivity [7]. Consequently, a difference in LST has been observed in city landscapes where core city space experiences higher LST than suburban/rural sites. This distinctive LST characteristic is defined by the surface urban heat island (SUHI). The SUHI has emerged because of the conversion of natural land into built-up space at the cost of water bodies, bare land, and forest [8,9].

LST intensification in urban setups is a perilous factor responsible for deteriorating urban climate and degrading human life and living [10,11]. Much attention is now given to mitigating its severity and threats to varied aspects of the environment by policymakers, health authorities, urban planners, climatologists, and environmentalists [12–14]. In the recent past, researchers have been engrossed in the interrelationship between LST and different land indices, such as the normalized difference built-up index (NDBI), enhanced built-up and bareness index (EBBI), normalized difference vegetation index (NDVI), and normalized difference moisture index (NDMI). Worldwide, scientists have been intensively focused on how, where, and what magnitude of land use/land cover (LULC) or land indices dynamics have been influencing the climatic conditions as a result of LST intensification and SUHI, which directly or indirectly make the environment uncomfortable and unhealthy for all animals and plants. Some studies rigorously found similar facts in Taipei city of Taiwan [15], Phoenix city of the United States of America (USA) [16], Singapore [10], Dhaka city of Bangladesh [17], Kathmandu valley of Nepal [18], Nanjing city of China [19], Beijing city of China [20], Tokyo city of Japan [21], Tehran city of Iran [13], 70 selected cities of Europe [22], Hong Kong [23], Baltimore–DC metropolitan area of the USA [24], and Cairo city of Egypt [25]. At the same time, researchers have discussed how their changing aspects affect and transform the environment of the city landscape in various Indian cities, such as Kolkata [26], Delhi and Mumbai [27], Chandigarh [28], Hyderabad [29], Noida [30], Lucknow [31], and Raipur [32].

Mal et al. (2020) conducted a study on the relationship of LST with LULC and elevation in the Ganga River basin, which includes major cities such as Kolkata, Patna, Allahabad (now Prayagraj), Varanasi, Lucknow, Kanpur, New Delhi, and Kathmandu during 2001–2019 using 1 km of MODIS Terra datasets [33]. Other studies over the Ganga River basin were also carried out for different cities such as Delhi [34], Kolkata [35], Kanpur and Patna [36], and Lucknow [30] using the MODIS/Landsat database. However, this study lacked a city-level analysis of LST profiling and SUHI information, especially for Prayagraj city. Furthermore, this study lacked effective land indices such as the NDBI, EBBI, NDVI, NDMI, normalized difference water index (NDWI), and soil-adjusted vegetation index (SAVI) to unfold the land dynamics and their role in the LST increase. Furthermore, the analytical results did not show details for thermal state analysis, including land indices dynamics and SUHI information, on a long spatiotemporal scale for Prayagraj city of India.

UN-Habitat (2018) introduced 17 sustainable development goals (SDGs). SDG-11 has stresses the city's resilience and sustainability and focuses on the significance of greenery and open spaces in bringing environmental viability and prosperity by coping with adverse local climate change and intense landscape transformation [37]. Various studies have shown that two main approaches help study the LST of urban climate, i.e., ground observations (GOBs) and satellite observations (SOBs). The GOB involves conventional data calculation of air temperature using urban and rural meteorological stations. The SOB has spatiotemporal resolutions with a mathematical background for estimating LST, and they are required to study the spatial variations of SUHI [3]. Therefore, in this study, we plan to use spatiotemporal Landsat imageries (1987–2018) to derive land indices, LST, and SUHI information in Prayagraj city using summer season (May–June) and winter season (December–January) datasets. We selected these times for our study on the basis of the finest spectral signature availability and albedo because of haze-free and cloud-free skies [38].

The primary aim of this present work is to examine the interrelations with LST dynamics using directional profiling on the summer/winter seasons during 1987–2018 in

Prayagraj city of India by investigating local level climate change through a long spatiotemporal analysis of six land indices (NDBI, EBBI, NDVI, NDMI, NDWI, and SAVI). In this connection, the tasks are (i) to assess the thermal state over the city landscape along with six land indices, (ii) to explore the interrelationship between six land indices and LST, (iii) to delineate the role of the six different land indices in LST increase using urban–rural directional profiling, (iv) to extract the SUHI state both at daytime and night-time, and (v) to extract the hotspots using Getis–Ord $G_i$* statistics. The dynamics of six land indices, LST, and their correlation are investigated to achieve these tasks. Then, the scenario of SUHI formation is discussed using directional profiling of LST by assessing how these six land indices impact the dynamics of LST in eight directions, namely, west, east, north, south, southwest, northeast, northwest, and southeast (center of the city to periphery). Next, the interrelationships between land indices and LST dynamics are validated using Google Earth images. Lastly, we delineate the hotspots to find warming or cooling spaces scattered over the city landscape.

## 2. Materials and Methods

### 2.1. Study Area

We selected Prayagraj city as a study area because this city was selected as a smart city by the MoHUA, i.e., the Ministry of Housing and Urban Affairs, Government of India (GoI), in 2015 [39]. This city is one of the biggest in terms of size, as well as historically enriched cities, in Uttar Pradesh state in India. Its location ranges from 25°23′7″N to 25°32′14″N latitude and 81°43′57″E to 81°53′59″E longitude, where the mean elevation is 93.77 m (Figure 1). The study area covers 72.98 km$^2$. This city is located over the holy place called Sangam (confluence of Ganga, Yamuna, and invisible Saraswati rivers) [40,41]. The sides of these rivers (Ganga and Yamuna) have been enriched by eroded materials from the Vidhyan uplands and Himalayas mountains [42]. The study area has a CWG-type climate, i.e., monsoon type with dry winters based on Koppen's climatic regions (KCR) scheme in India [43], and it has 744.1 mm of mean annual rainfall and 20–32.6 °C of mean annual temperature [44]. This city is widely known for religious gatherings at 6 year intervals, Maha Kumbh Mela and Ardha Kumbh Mela, where >100 million pilgrims congregate to make it the largest congregation in Asia [40].

### 2.2. Data Used

This study uses Landsat 5 (TM) and Landsat 8 (OLI/TIRS) satellite imageries with a spatial resolution of 30 m. These are employed for four distinct time points for distinctive seasons in summer and winter. The summer time points (STP) are (i) 4 June 1988 (S1) of Landsat 5 (TM), (ii) 12 May 1997 (S2) of Landsat 5 (TM), (iii) 10 May 2008 (S3) of Landsat 5 (TM), and (iv) 22 May 2018 (S4) of Landsat 8 (OLI/ TIRS). The winter time points (WTP) are (i) 11 December 1987 (W1) of Landsat 5 (TM), (ii) 3 December 1996 (W2) of Landsat 5 (TM), (iii) 16 January 2007 (W3) of Landsat 5 (TM), and (iv) 16 December 2018 (W4) of Landsat 8 (OLI/TIRS). We selected about 10 years of the gap to depict the dynamics of LST and land indices seasonally (summer and winter).

This study uses night-time LST derived from MODIS (Terra) satellite datasets to investigate summer/winter seasonal LST dynamics and SUHI state at night-times from 2007 to 2018 only as this satellite has been providing imagery since 2000. Before 2000, there were no available data on night-time LST. Table 1 shows the satellite datasets used in this study. The software, ERDAS IMAGINE 2014 was used for preprocessing these satellite images. The same dry summer and dry winter seasons were selected for obtaining the cloud-free data with the finest spectral information.

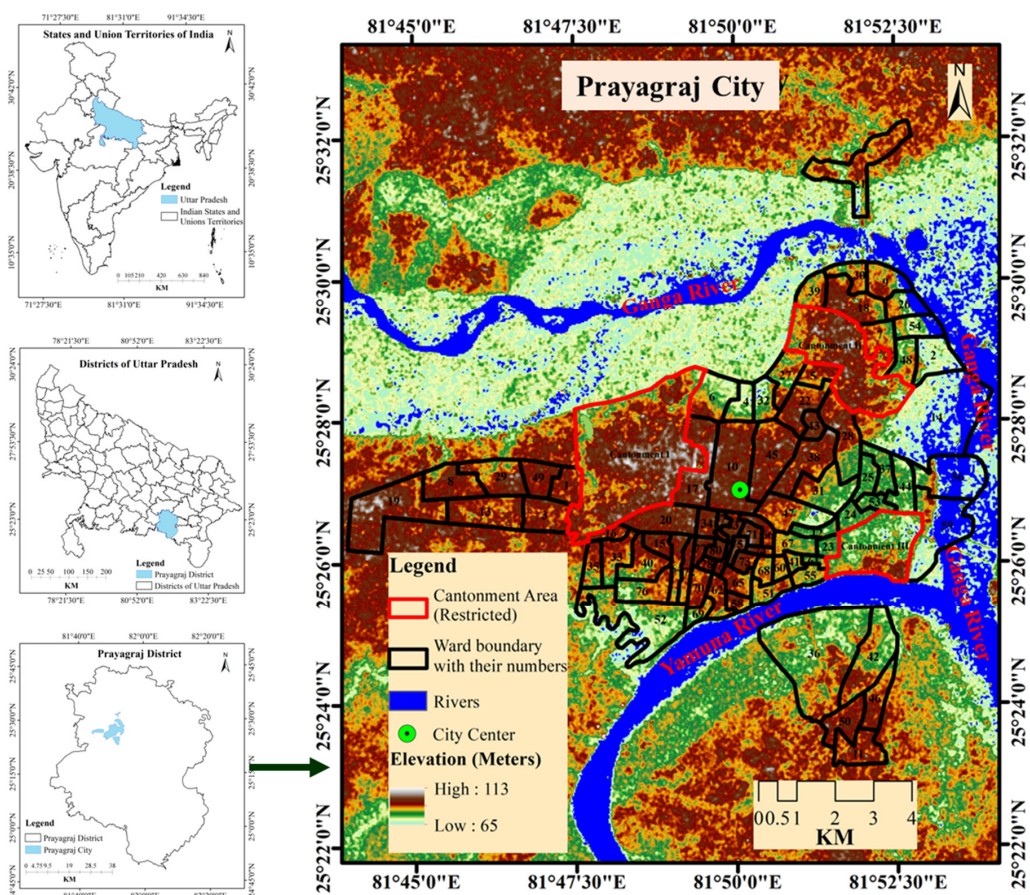

**Figure 1.** Map of Prayagraj city (India).

**Table 1.** Summary of the data used.

| Satellite (Sensor)/Ancillary Data | Path/Row | Resolution/Scale | Season | Acquisition Date | Time (GMT) | Constants of Thermal Conversion | | Source |
|---|---|---|---|---|---|---|---|---|
| | | | | | | $K_1$ | $K_2$ | |
| Landsat-5 (TM) | 143/42 | 30 m | Summer | 04-06-1988 | 04:31:36 | 607.76 (Band 6) | 1260.56 (Band 6) | United States Geological Survey (USGS) web portal (https://earthexplorer.usgs.gov/, accessed on 15 January 2019) |
| | | | | 12-05-1997 | 04:29:21 | 607.76 (Band 6) | 1260.56 (Band 6) | |
| | | | | 10-05-2008 | 04:49:45 | 607.76 (Band 6) | 1260.56 (Band 6) | |
| Landsat-8 (OLI/TIRS) | | | | 22-05-2018 | 05:00:01 | 774.8853 (Band 10) | 1321.0789 (Band 10) | |
| Landsat-5 (TM) | 143/42 | 30 m | Winter | 11-12-1987 | 04:29:34 | 607.76 (Band 6) | 1260.56 (Band 6) | |
| | | | | 03-12-1996 | 04:22:45 | 607.76 (Band 6) | 1260.56 (Band 6) | |
| | | | | 16-01-2007 | 04:55:59 | 607.76 (Band 6) | 1260.56 (Band 6) | |
| Landsat-8 (OLI/TIRS) | | | | 16-12-2018 | 05:00:58 | 480.8883 (Band 11) | 1201.1442 (Band 11) | |
| MODIS (Terra) | - | 1 km | Summer | 11-05-2008 | Night–time | - | - | |
| | - | | | 22-05-2018 | | - | - | |
| | - | 1 km | Winter | 17-01-2007 | | - | - | |
| | - | | | 15-12-2018 | | - | - | |
| ASTER | - | 30 m | - | 13-09-2017 | - | - | - | |
| Ward boundary map | - | 1:21,600 | - | - | - | - | - | Prayagraj Nagar Nigam |
| Political map | - | 1:4 M | - | 2014 | - | - | - | Survey of India |

### 2.3. Methods

Figure 2 shows the overall methodological framework for the execution process. All the methods were discussed under subsequent heads.

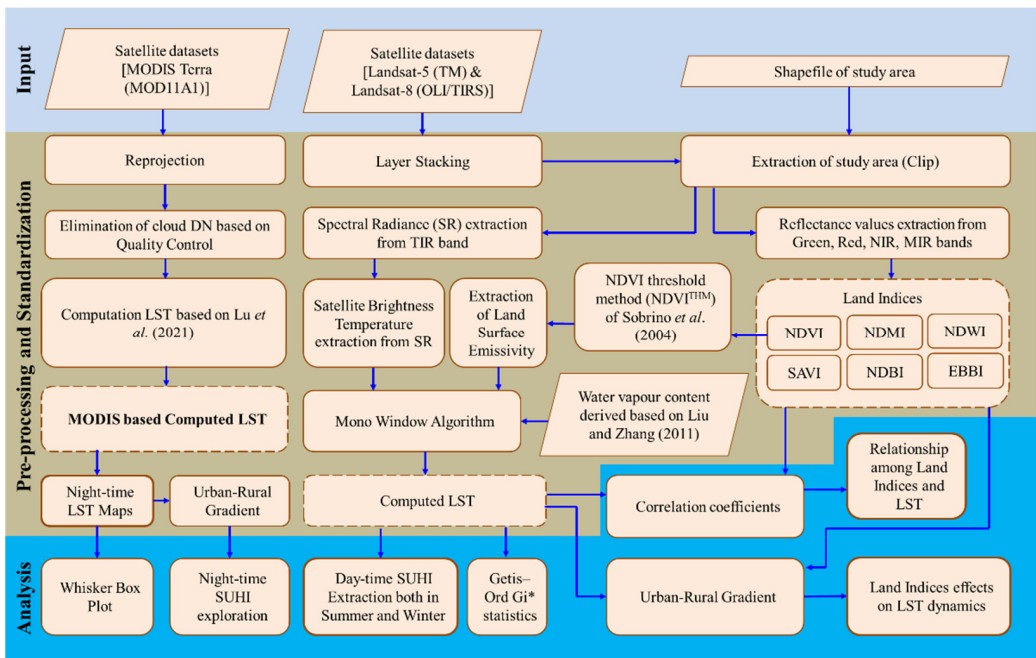

**Figure 2.** Overall framework of the present research procedure [23,45,46].

### 2.3.1. Land Indices
NDBI

In the evaluation of urban climate, NDBI is an important indicator. It ranges from $-1$ to $+1$. A higher positive value specifies bare soils. Lower positive and negative values adjacent to 0 show the dominance of built-up space. Higher negative values specify vegetation and water bodies. NDBI is calculated using Equation (1).

$$\text{NDBI} = \left[\frac{\text{MIR}_{\text{Band}} - \text{NIR}_{\text{Band}}}{\text{MIR}_{\text{Band}} + \text{NIR}_{\text{Band}}}\right], \tag{1}$$

where, in Landsat 5 TM, $\text{NIR}_{\text{Band}}$ is band 4, and $\text{MIR}_{\text{Band}}$ is band 5, whereas, in Landsat 8, $\text{NIR}_{\text{Band}}$ is band 5 and $\text{MIR}_{\text{Band}}$ is band 6.

EBBI

EBBI is widely used as a significant indicator in assessing urban climate. It ranges from 0 to $+1$. A higher positive value (perceived threshold > 0.1) specifies bare soils. A lower positive value (perceived threshold between 0.06 and 0.1) specifies built-up space. m positive value (perceived threshold < 0.06) adjacent to 0 specifies water bodies and vegetation. EBBI is calculated using Equation (2) [17].

$$\text{EBBI} = \left[\frac{\text{MIR}_{\text{Band}} - \text{NIR}_{\text{Band}}}{10\sqrt{\text{MIR}_{\text{Band}} + \text{TIR}_{\text{Band}}}}\right], \tag{2}$$

where, in Landsat 5 TM, $\text{NIR}_{\text{Band}}$ is band 4, $\text{MIR}_{\text{Band}}$ is band 5, and $\text{TIR}_{\text{Band}}$ is band 6, whereas, in Landsat 8, $\text{NIR}_{\text{Band}}$ is band 5, $\text{MIR}_{\text{Band}}$ is band 6, and $\text{TIR}_{\text{Band}}$ is band 10.

NDMI

Another critical indicator in urban climate assessment is NDMI. It ranges from $-1$ to $+1$. A positive value specifies vegetation and water bodies. A negative value indicates bare

soils and built-up areas. It provides information on the moisture present in the landscape. NDMI is calculated using Equation (3) [47].

$$\text{NDMI} = \left[\frac{\text{NIR}_{\text{Band}} - \text{MIR}_{\text{Band}}}{\text{NIR}_{\text{Band}} + \text{MIR}_{\text{Band}}}\right] \tag{3}$$

where, in Landsat 5 TM, $\text{NIR}_{\text{Band}}$ is band 4, and $\text{MIR}_{\text{Band}}$ is band 5, whereas, in Landsat 8, $\text{NIR}_{\text{Band}}$ is band 5, and $\text{MIR}_{\text{Band}}$ is band 6.

NDVI

NDVI ranges from −1 to +1. A higher positive value specifies vegetation. A lower positive value indicates bare soils or built-up areas. The negative values specify water bodies. NDVI is calculated using Equation (4) [12].

$$\text{NDVI} = \left[\frac{\text{NIR}_{\text{Band}} - \text{Red}_{\text{Band}}}{\text{NIR}_{\text{Band}} + \text{Red}_{\text{Band}}}\right] \tag{4}$$

where Landsat 5 TM defines $\text{NIR}_{\text{Band}}$ as band 4 and $\text{Red}_{\text{Band}}$ as band 3, whereas Landsat 8 OLI/TIRS defines $\text{NIR}_{\text{Band}}$ as band 5 and $\text{Red}_{\text{Band}}$ as band 4.

NDWI

NDWI ranges from −1 to +1. A positive value specifies water bodies. A lower positive value adjacent to 0 specifies vegetation space. Negative values specify bare soils and built-up areas. NDWI is calculated using Equation (5) [48].

$$\text{NDWI} = \left[\frac{\text{Green}_{\text{Band}} - \text{NIR}_{\text{Band}}}{\text{Green}_{\text{Band}} + \text{NIR}_{\text{Band}}}\right] \tag{5}$$

where, in Landsat 5 TM, $\text{Green}_{\text{Band}}$ is band 2, and $\text{NIR}_{\text{Band}}$ is band 4, whereas, in Landsat 8, $\text{Green}_{\text{Band}}$ is band 3, and $\text{NIR}_{\text{Band}}$ is band 5.

SAVI

SAVI ranges from −1 to +1. A positive value specifies vegetation. A lower positive value adjacent to 0 specifies water bodies. Negative values specify bare soils and built-up space. It is calculated using Equation (6) [49].

$$\text{SAVI} = \left[\frac{\text{NIR}_{\text{Band}} - \text{Red}_{\text{Band}}}{\text{NIR}_{\text{Band}} + \text{Red}_{\text{Band}} + L} \times (L+1)\right] \tag{6}$$

where Landsat 5 TM defines $\text{NIR}_{\text{Band}}$ as band 4 and $\text{Red}_{\text{Band}}$ as band 3, whereas Landsat 8 OLI/TIRS defines $\text{NIR}_{\text{Band}}$ as band 5 and $\text{Red}_{\text{Band}}$ as band 4. $L$ is the soil brightness correction factor, and it is a constant 0.5.

2.3.2. LST Retrieval

Landsat-Based LST Calculation

Several algorithms are available for retrieving LST for distinct satellite sensors. The most popular algorithms are the mono–window algorithm (MWA) [50], radiative transfer equation (RTE) algorithm [51], split-window algorithm (SWA) [52–54], and single-channel algorithm (SCA) [55,56] for retrieving LST using thermal bands of Landsat. RTE cannot be considered for use if the information of the atmospheric profile is not available on in situ parameters at the satellite pass [57]. The SWA provides accurate results but was not selected for use in the study area as it is specific to band 10 of Landsat 8 (OLI/TIRS) data only for LST computation because of its better calibration. The MWA and SCA also give good results [33,58]. In this study, we selected MWA for LST computation from multitemporal Landsat images because MWA shows significant accuracy for computing LST, with three essential parameters being indispensable to LST retrieval, i.e., the ground

emissivity, the atmospheric transmittance (AT), and the effective mean temperature of the atmosphere [59], calculated using Equations (7)–(9), respectively.

$$T_s = \frac{\{a(1 - C - D) + [b(1 - C - D) + C + D] \times T_b - D \times T_a\}}{C} \tag{7}$$

$$C = \varepsilon \times \tau \tag{8}$$

$$D = (1 - \tau) \times [1 + (1 - \varepsilon)\tau] \tag{9}$$

where $T_s$ defines the LST (K), $T_a$ defines the mean atmospheric temperature (K), $T_b$ defines the at-sensor pixel brightness temperature (K), $C$ and $D$ algorithm parameters are estimated through land surface emissivity (LSE) and AT, $\varepsilon$ defines LSE, $\tau$ defines AT, and $a$ and $b$ are constants of the algorithm ($-67.355351$ and $0.458606$, respectively). The following steps are needed to calculate Equations (7)–(9).

*Step 1*: The TIR band's pixels are converted into radiance. The radiance is computed using Equation (10) for band 6 of Landsat 5 (TM) and Equation (11) for band 10 of Landsat 8 (OLI/TIRS).

$$L_\lambda = \Lambda \times QCAL + \Gamma \tag{10}$$

$$L_\lambda = \frac{L_{\max} - L_{\min}}{QCAL_{\max} - QCAL_{\min}} \times (QCAL - QCAL_{\min}) - L_{\min} \tag{11}$$

where $L_\lambda$ represents spectral radiance at the top of atmosphere (TOA) (W/(m$^2$·sr·μm)), $\Lambda$ represents the multiplicative rescaling factor for each specific band in the metadata, $\Gamma$ represents the additive rescaling factor for each specific band in the metadata, and $QCAL$ represents the quantized and calibrated digital number (DN) values of standard product. $QCAL_{\max}$ and $QCAL_{\min}$ represent the maximum and minimum DN values of the images, respectively. $L_{\max}$ and $L_{\min}$ represent the TIR band's spectral radiance at $QCAL_{\max}$ and $QCAL_{\min}$, respectively. These values of the rescaling factor are available in the metadata of respective Landsat images.

Then, the at-sensor brightness temperature (BT) is computed using Equation (12).

$$T_b = \left[ \frac{K_2}{\ln\left(\frac{K_1}{L_\lambda} + 1\right)} \right] \tag{12}$$

where $T_b$ represents the at-sensor BT (K), and $K_1$ and $K_2$ (Wm$^{-2}$) are thermal conversion constants (prelaunch calibration) mentioned in the metadata (Table 1) of the respective sensors of Landsat datasets.

*Step 2*: To compute LST, LSE is one of the indispensable parameters [60]. The NDVI threshold (NDVI$^{\text{THR}}$) method was selected to calculate the LSE because of its significance in segregating pixels of vegetation, water, and soil [45]. LSE can be computed using Equations (13)–(15).

$$\varepsilon = \varepsilon_v P_v + \varepsilon_s (1 - P_V) + C \tag{13}$$

$$P_v = \left[ \frac{NDVI - NDVI_S}{NDVI_V - NDVI_S} \right]^2 \tag{14}$$

$$C = (1 - \varepsilon_s)\varepsilon_v F(1 - P_V) \tag{15}$$

where $\varepsilon$ represents LSE, $\varepsilon_v$ represents vegetation emissivity, $\varepsilon_s$ represents soil emissivity, $P_V$ represents proportionate of vegetation, $C$ represents constant of surface characteristics, $NDVI_s$ represents NDVI of pure soil, $NDVI_v$ represents NDVI of pure vegetation, and $F$ represents a geometric factor commonly considered as 0.55 [45,61].

The constant values of LSE were calculated using Equation (16) for Landsat 5 (TM) and Equation (17) for Landsat 8 (OLI/TIRS) [61], where $\rho_{\text{Red}}$ represents the reflectance value of respective red bands of the imageries.

$$\varepsilon = \begin{cases} 0.979 + 0.035\rho_{\text{Red}} \rightarrow NDVI < 0.2 \\ 0.004P_V + 0.986\rho_{\text{Red}} \rightarrow 0.2 \leq NDVI \geq 0.5 \\ 0.99 \rightarrow NDVI > 0.5 \end{cases} \tag{16}$$

$$\varepsilon = \begin{cases} 0.979 + 0.046\rho_{\text{Red}} \rightarrow NDVI < 0.2 \\ 0.989P_V + 0.977\rho_{\text{Red}} \rightarrow 0.2 \leq NDVI \geq 0.5 \\ 0.987 + C \rightarrow NDVI > 0.5 \end{cases} \tag{17}$$

*Step 3*: The AT is another indispensable parameter to calculate LST. Before calculating AT, water vapor content should be calculated on the basis of the atmospheric profile (Table 2) using Equation (18) [59,61]. Then, AT is computed using Equation (19).

$$w = 0.0981 \times \left[ 10 \times 0.6108 \times \exp\left( \frac{17.27 \times (T_0 - 273.15)}{237.3 + (T_0 - 273.15)} \right) \times RH \right] + 0.1697 \tag{18}$$

where $w$ represents the water vapor content (g/cm$^2$), $RH$ represents the relative humidity, and $T_0$ represents the near-surface temperature. These atmospheric parameters ($RH$ and $T_0$) were obtained from the Prediction of Worldwide Energy Resource (POWER) Project of the National Aeronautics and Space Administration (NASA) (https://power.larc.nasa.gov/, accessed on 27 November 2022).

$$\tau = 1.031412 - 0.11536w \tag{19}$$

**Table 2.** Estimation equations of atmospheric transmittance.

| Atmospheric Profile | Water Vapor ($w$) (g/cm$^2$) | Equation for Transmittance Estimation | Squared Correlation ($R^2$) | Standard Error |
|---|---|---|---|---|
| High air temperature (summer) | 0.4–1.6 | $\tau = 0.974290 - 0.08007w$ | 0.99611 | 0.002368 |
| High air temperature (summer) | 1.6–3.0 | $\tau = 1.031412 - 0.11536w$ | 0.99827 | 0.002539 |
| Low air temperature (winter) | 0.4–1.6 | $\tau = 0.982007 - 0.09611w$ | 0.99463 | 0.003340 |
| Low air temperature (winter) | 1.6–3.0 | $\tau = 1.053710 - 0.14142w$ | 0.99899 | 0.002375 |

*Step 4*: The effective mean atmospheric temperature is another indispensable parameter for computing LST (Table 3). It can be computed using Equation (20).

$$T_a = 16.0110 + 0.92621T_0 \tag{20}$$

**Table 3.** The estimation equation for effective mean atmospheric temperature in four standard atmospheres.

| Standard Atmosphere | Estimation Equation (Kelvin) |
|---|---|
| For USA 1976 | $T_a = 25.9396 + 0.88045T_0$ |
| For tropical | $T_a = 17.9769 + 0.91715T_0$ |
| For mid-latitude summer | $T_a = 16.0110 + 0.92621T_0$ |
| For mid-latitude winter | $T_a = 19.2704 + 0.91118T_0$ |

Lastly, Equation (21) is applied to get LST in degrees Celsius, where $T_s(Kelvin)$ is converted into $T_s(°C)$.

$$T_s(°C) = T_s(Kelvin) - 273.15 \tag{21}$$

Furthermore, we validated the LST and weather information with NASA's POWER project over selected sample location (Latitude: 25.4495 and Longitude: 81.8417) of selected

time points between 1987 and 2018 for this study area, Prayagraj city, which are available in an excel sheet provided as a Supplementary Materials.

MODIS-Based Night-Time LST Calculation

Night-time LST was retrieved using MODIS night-time datasets. First of all, the MODIS night-time dataset reprojection was changed. Then, cloud-affected areas were eliminated using preprocessed quality control. Each pixel DN value was then converted into LST (°C) using Equation (22) [46].

$$T_s(°C) = DN \times 0.02 - 273.15 \qquad (22)$$

2.3.3. Influence of Land Indices on LST

Pearson's correlation coefficient (*r*)-based analysis was incorporated to assess the distinct effect of land indices (NDBI, EBBI, NDVI, NDMI, NDWI, and SAVI) on the intensification of LST. Accordingly, scatter plots were prepared for all four distinct summer time points, i.e., S1, S2, S3, and S4, and all four distinct winter time points, i.e., W1, W2, W3, and W4. In this analysis, '*r*' represents the relationships, i.e., LST vs. NDBI, LST vs. EBBI, LST vs. NDMI, LST vs. NDVI, LST vs. NDWI, and LST vs. SAVI, where the dependent variable is the LST, and the independent variables are land indices (NDBI, EBBI, NDVI, NDMI, NDWI, and SAVI). Pearson's '*r*' is calculated using Equation (23) [62,63].

$$r = \frac{\sum_{i}^{n}(x_i - \bar{x})(y_i - \bar{y})}{\sqrt{\sum_{i=1}^{n}(x_i - \bar{x})^2}\sqrt{\sum_{i=1}^{n}(y_i - \bar{y})^2}} \qquad (23)$$

where $x_i$ defines the values of land indices (NDBI, EBBI, NDVI, NDMI, NDWI, and SAVI), and $y_i$ defines the LST values.

2.3.4. Intensity of SUHI Calculation

The SUHI is defined by the observed difference of LST between the urban space and the suburban/rural space over a city landscape. It was previously well defined by Oke and East [64] and Oke [65] using Equation (24), which is very popular in the literature [9,13].

$$T_{U-R} = T_U - T_R \qquad (24)$$

where $T_{U-R}$ is the intensity of the SUHI, $T_U$ represents the LST of urban space, and $T_R$ represents the LST of suburban/rural space.

2.3.5. Hotspot Analysis (Getis–Ord $G_i$*)

The spatial LST distribution over the city landscape was examined using hotspot analysis (Getis–Ord $G_i$*) to characterize both hot and cold spots over the city using each feature (LST value) on the basis of its neighboring features. Hotspots are the clustered areas of high values of the feature, whereas cold spots are the clustered areas of low values of the feature. The Getis–Ord $G_i$* statistic is derived from Equations (25)–(27) [66].

$$G_i^* = \frac{\sum\limits_{j=1}^{n} w_{i,j}x_j - \overline{X}\sum\limits_{j=1}^{n} w_{i,j}}{S\sqrt{\dfrac{n\sum\limits_{j=1}^{n} w_{i,j}^2 - \left(\sum\limits_{j=1}^{n} w_{i,j}\right)^2}{n-1}}} \qquad (25)$$

$$\overline{X} = \frac{\sum\limits_{j=1}^{n} x_j}{n} \qquad (26)$$

$$S = \sqrt{\frac{\sum\limits_{j=1}^{n} x_j^2}{n} - \overline{X}^2} \tag{27}$$

where $x_j$ is defined by the feature attribute value of $j$, $w_{i,j}$ is defined by the spatial weight between features $i$ and $j$, and $n$ is defined by the total number of features.

## 3. Results

### 3.1. Seasonal Spatiotemporal LST Dynamics

The summer spatiotemporal distribution of LST dynamics for Prayagraj city is shown in Figure 3a for summer time points S1, S2, S3, and S4 with boxplots in Figure 3b and their statistics in Table 4. The mean LST witnessed was 38.20 °C in S1, which increased to 40.44 °C in S2, but declined to 37.49 °C in S3, before again inclining to 38.09 °C in S4. In S1, the foremost area of warm temperature was northeast, and that of cool temperature was southeast. In S2, the foremost area of warm temperature was west (except barren land in the northwest at 8–8.5 km), and that of cool temperature was southeast (except for the area of Ganga River flow in the east at 8–9 km). In S3, the foremost area of warm temperature was southwest, and that of cool temperature was northwest (except for the area of the Ganga River flow in the northeast at 8–9 km). However, in S4, the foremost area of warm temperature was the northeast. The foremost area of cool temperature was the northwest (except for vegetation coverage area in the northeast at 1.5–2.5 km). We recommend further research on the surface types or land use/land cover (LULC) classes of the city landscape in Sarif and Gupta (2022) to additionally determine their distribution over the city landscape [67].

**Table 4.** Summer/winter seasonal LST dynamics of Prayagraj city (1987–2018).

| *Summer LST Dynamics* | | | | |
|---|---|---|---|---|
| **Date** | **Minimum (°C)** | **Maximum (°C)** | **Mean (°C)** | **Standard Deviation** |
| 04-06-1988 | 29.72 | 42.90 | 38.20 | 1.83 |
| 12-05-1997 | 26.67 | 46.66 | 40.44 | 2.47 |
| 10-05-2008 | 27.52 | 44.85 | 37.49 | 2.15 |
| 22-05-2018 | 30.84 | 44.21 | 38.09 | 1.66 |
| *Winter LST Dynamics* | | | | |
| **Date** | **Minimum (°C)** | **Maximum (°C)** | **Mean (°C)** | **Standard Deviation** |
| 11-12-1987 | 13.26 | 24.11 | 19.72 | 1.14 |
| 03-12-1996 | 13.80 | 23.68 | 19.41 | 1.15 |
| 16-01-2007 | 13.31 | 24.11 | 18.06 | 1.47 |
| 16-12-2018 | 13.77 | 25.11 | 19.84 | 1.24 |

The winter spatiotemporal LST is mapped in Figure 4a showing the distribution of LST dynamics. Their statistics are presented in Table 4 and Figure 4b. The mean LST witnessed was 19.72 °C in W1, which declined to 19.41 °C in W2. It further declined to 18.06 °C in W3. However, again, it inclined to 19.84 °C in W4. In W1, the foremost area of warm temperature was the northwest, and that of cool temperature was North (except for the area of the Ganga River flow in the northeast at 8–9 km). In W2, the foremost area of warm temperature again was the northwest, and that of cool temperature was the east (except for the area of the Ganga River flow in the northeast at 8–9 km). In W3, the foremost area of warm temperature was the northwest, and that of cool temperature was the northeast. In W4, the foremost area of warm temperature was the southeast, and that of cool temperature was the northwest (except for the area of vegetation coverage in the northeast at 1.5–2.5 km).

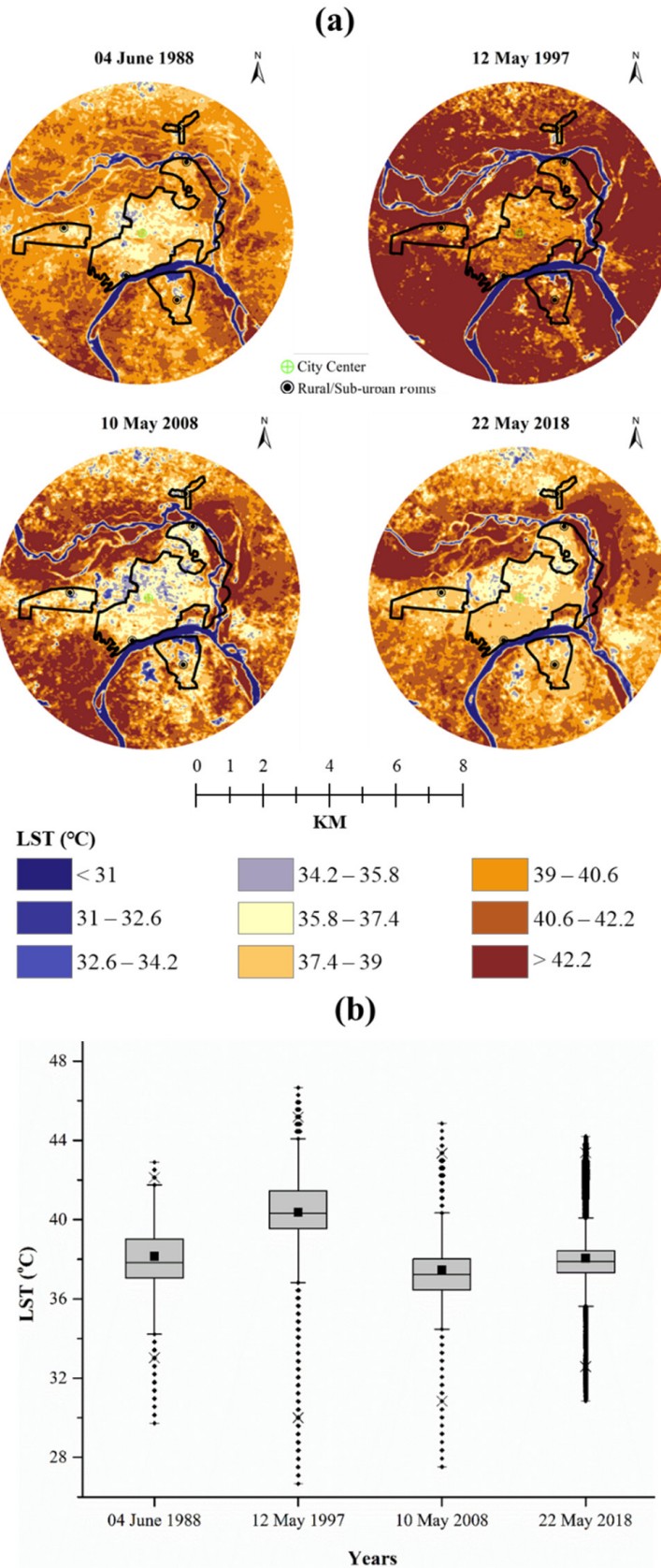

**Figure 3.** Spatiotemporal dynamics of LST in the summer season over Prayagraj city: (**a**) LST maps during 1988–2018 (five rural/suburban areas shown here used for computing SUHI) and (**b**) boxplots of LST dynamics during 1988–2018.

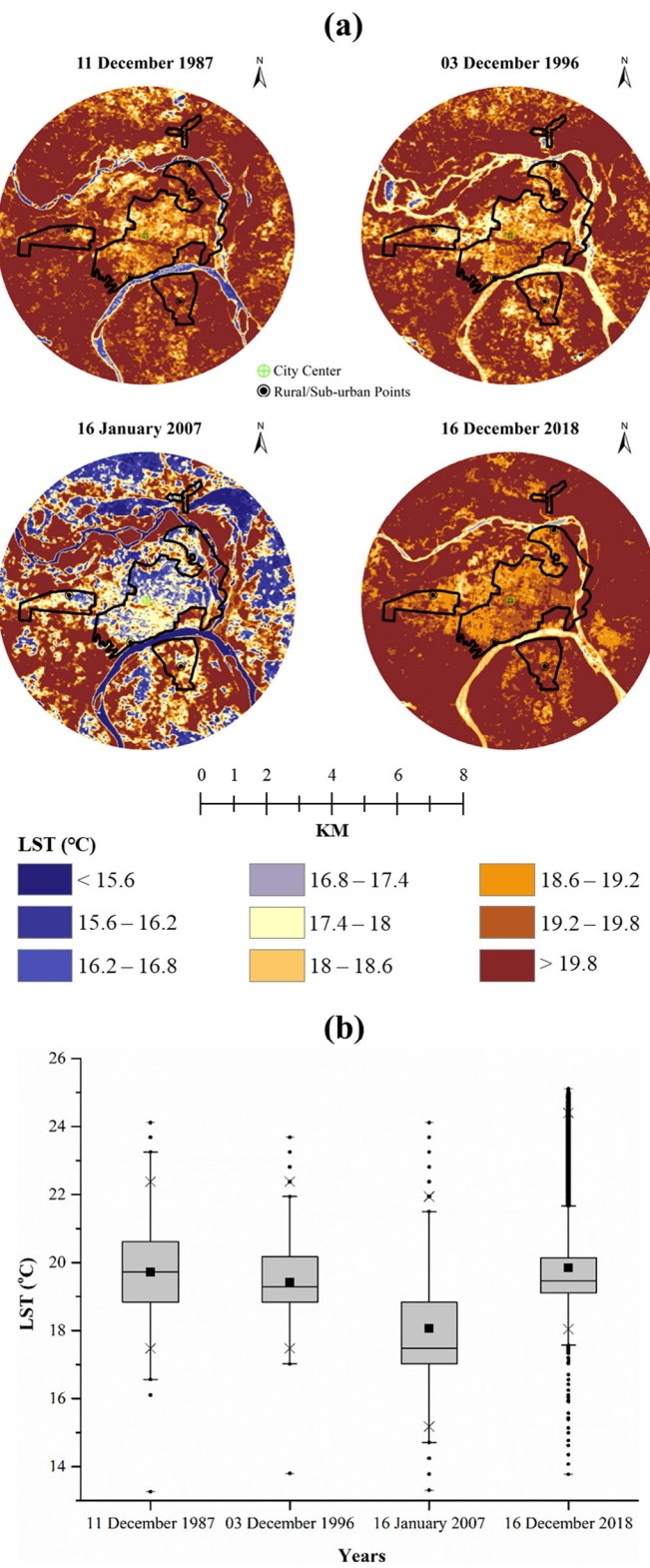

**Figure 4.** Spatiotemporal dynamics of LST (°C) in winter season over Prayagraj city: (**a**) LST maps during 1987–2018 (five rural/suburban areas shown here used for computing SUHI) and (**b**) boxplots of LST dynamics during 1987–2018.

### 3.2. Seasonal Magnitude of LST Based on Multiple Ring Profiling

The magnitude of the summer mean LST difference between the periods based on multiple ring profiling was extracted at 0.5 km intervals from the city's center to the city's

periphery (Table 5 and Figure 5). It was detected that each zone at each period witnessed a higher temperature in comparison to its preceding time points (except for the period between S2 and S3) by a significant amount. Substantial temperature intensification was experienced in each distinct zone from the city center to the periphery. However, a declining trend was observed in the period of S1–S2.

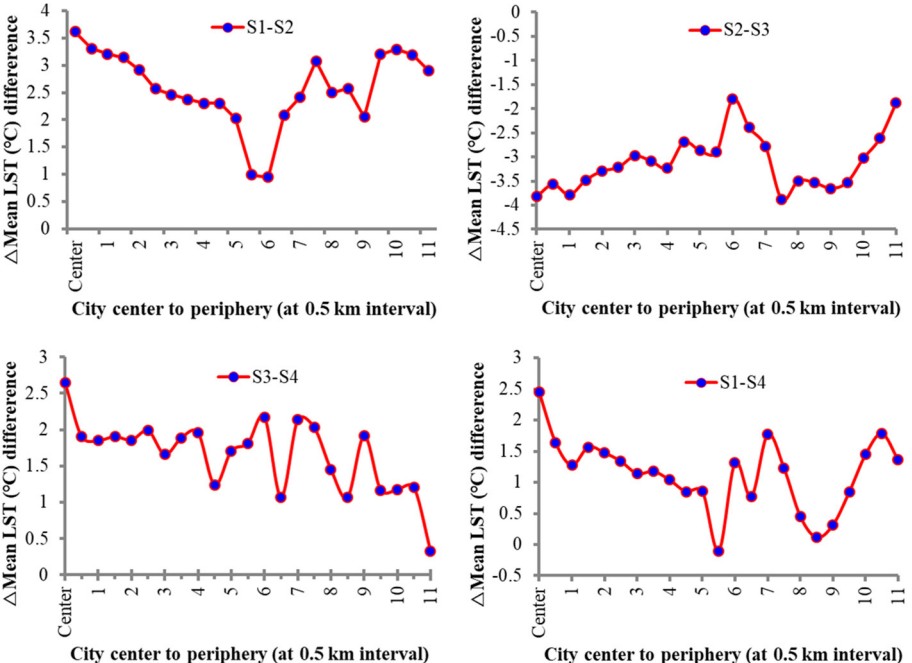

**Figure 5.** Magnitude of the summer mean LST in different periods (1988–2018).

**Between S1 and S2**, the mean LST observed amplifying pattern ranged from 0.94 °C to 3.62 °C, wherein the lowest intensified zone was 12 (5.5–6 km) by 0.94 °C because there was existence of grass and forest land. The highest intensified zone was 0 (city center) by 3.62 °C because of the presence of highly dense, impervious land. **Between S2 and S3**, the difference in mean LST observed a declining pattern which ranged from −1.80 °C to −3.88 °C, wherein the lowest decline zone was 12 (5.5–6 km) by −1.80 °C because of the dominance of grass and forest land, but the highest declined zone was 15 (7–7.5 km) by −3.88 °C because of the dominance of sand and barren land. **Between S3 and S4**, the difference in mean LST was found in an amplifying pattern that ranged from 0.33 °C to 2.65 °C, wherein the lowest intensified zone was 22 (10.5–11 km) by 0.33 °C because of the presence of grassland. The highest intensified zone was again 0 (city center) by 2.65 °C because of the presence of highly dense, impervious land. **Between S1 and S4**, the difference in mean LST showed an increasing pattern which ranged from 0.12 °C to 2.45 °C except for zone 11 (5–5.5 km), in which the magnitude of mean LST declined by −0.10 °C because of forest and grassland, wherein the lowest intensified zone was 17 (8–8.5 km) by 0.12 °C because of the presence of the Ganga River flow. The highest intensified zone again was 0 (city center) by 2.45 °C because of the presence of highly dense, impervious land.

The magnitude of the winter mean LST difference distributional pattern between distinctive periods on the basis of multiple ring profiling was also extracted at 0.5 km intervals from the center of the city to the periphery of the city (Table 5 and Figure 6). It was detected that each zone at each period witnessed lower temperatures in comparison to its preceding time points by a significant amount except for the period between W3 and W4. This means that a substantial temperature reduction was experienced in each zone from the city center to the periphery. However, an amplifying trend was observed during the period of W3–W4.

**Table 5.** Periodical LST magnitudes based on multiple ring buffers from the city center to the periphery at 0.5 km of intervals in Prayagraj city (1987–2018).

| Zones | Distance from the City Center at 0.5 km of Interval | Periodical Difference of Mean LST (°C) | | | |
|---|---|---|---|---|---|
| | **Summer Magnitude** | **S1–S2** | **S2–S3** | **S3–S4** | **S1–S4** |
| 0 | Center | 3.62 | −3.82 | 2.65 | 2.45 |
| 1 | 0.5 | 3.31 | −3.57 | 1.90 | 1.64 |
| 2 | 1 | 3.21 | −3.78 | 1.85 | 1.28 |
| 3 | 1.5 | 3.14 | −3.48 | 1.91 | 1.57 |
| 4 | 2 | 2.91 | −3.29 | 1.86 | 1.48 |
| 5 | 2.5 | 2.57 | −3.21 | 1.99 | 1.34 |
| 6 | 3 | 2.46 | −2.98 | 1.66 | 1.14 |
| 7 | 3.5 | 2.37 | −3.09 | 1.89 | 1.18 |
| 8 | 4 | 2.30 | −3.22 | 1.96 | 1.04 |
| 9 | 4.5 | 2.30 | −2.69 | 1.24 | 0.85 |
| 10 | 5 | 2.02 | −2.87 | 1.71 | 0.86 |
| 11 | 5.5 | 0.99 | −2.90 | 1.81 | −0.10 |
| 12 | 6 | 0.94 | −1.80 | 2.18 | 1.32 |
| 13 | 6.5 | 2.09 | −2.39 | 1.07 | 0.77 |
| 14 | 7 | 2.42 | −2.78 | 2.14 | 1.77 |
| 15 | 7.5 | 3.08 | −3.88 | 2.03 | 1.23 |
| 16 | 8 | 2.50 | −3.50 | 1.46 | 0.45 |
| 17 | 8.5 | 2.58 | −3.53 | 1.07 | 0.12 |
| 18 | 9 | 2.05 | −3.65 | 1.92 | 0.32 |
| 19 | 9.5 | 3.20 | −3.52 | 1.17 | 0.84 |
| 20 | 10 | 3.30 | −3.03 | 1.18 | 1.45 |
| 21 | 10.5 | 3.20 | −2.61 | 1.21 | 1.79 |
| 22 | 11 | 2.91 | −1.88 | 0.33 | 1.36 |
| | **Winter Magnitude** | **W1–W2** | **W2–W3** | **W3–W4** | **W1–W4** |
| 0 | Center | 0.45 | −2.24 | 2.44 | 0.64 |
| 1 | 0.5 | 0.22 | −1.49 | 1.89 | 0.62 |
| 2 | 1 | −0.03 | −1.62 | 1.88 | 0.22 |
| 3 | 1.5 | −0.05 | −1.47 | 1.83 | 0.31 |
| 4 | 2 | −0.09 | −1.59 | 1.96 | 0.29 |
| 5 | 2.5 | −0.15 | −1.74 | 2.09 | 0.20 |
| 6 | 3 | −0.16 | −1.61 | 1.87 | 0.10 |
| 7 | 3.5 | −0.25 | −1.72 | 2.00 | 0.03 |
| 8 | 4 | −0.03 | −1.71 | 1.76 | 0.02 |
| 9 | 4.5 | −0.54 | −0.98 | 1.15 | −0.36 |
| 10 | 5 | −0.17 | −1.20 | 1.72 | 0.34 |
| 11 | 5.5 | −0.76 | −1.34 | 1.88 | −0.21 |
| 12 | 6 | −0.49 | −1.38 | 2.45 | 0.58 |
| 13 | 6.5 | −0.20 | −0.78 | 1.41 | 0.42 |
| 14 | 7 | −0.28 | −1.04 | 1.81 | 0.49 |
| 15 | 7.5 | −0.71 | −1.04 | 1.20 | −0.55 |
| 16 | 8 | −0.68 | −0.90 | 1.17 | −0.42 |
| 17 | 8.5 | −0.51 | −1.02 | 0.64 | −0.89 |
| 18 | 9 | −0.70 | −1.33 | 1.35 | −0.67 |
| 19 | 9.5 | −0.65 | −0.86 | 0.95 | −0.56 |
| 20 | 10 | −0.36 | −0.48 | 1.11 | 0.27 |
| 21 | 10.5 | −0.53 | −1.07 | 1.99 | 0.38 |
| 22 | 11 | −0.28 | −1.14 | 2.48 | 1.06 |

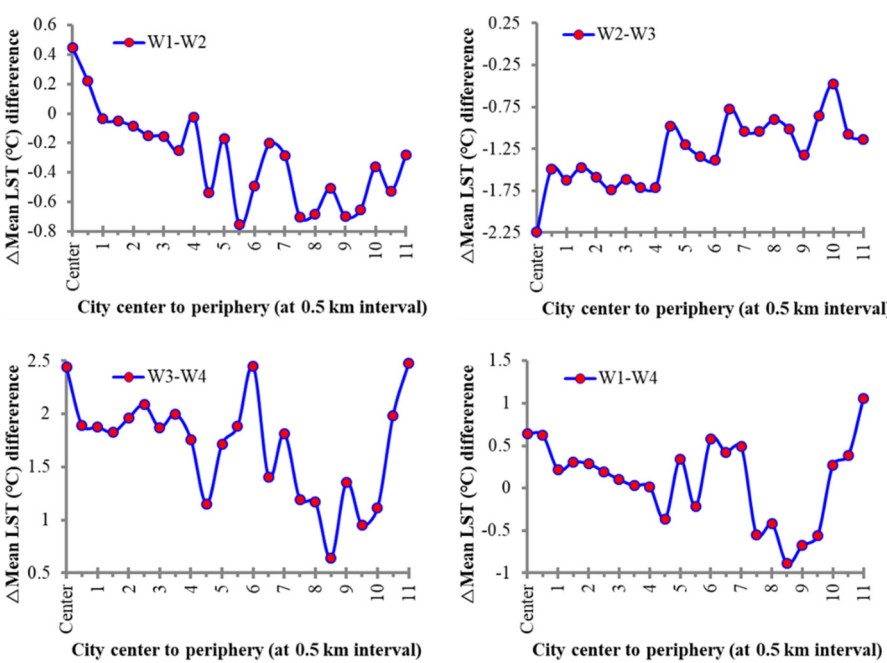

**Figure 6.** Magnitude of the winter mean LST in different periods (1987–2018).

**Between W1 and W2**, the difference in mean LST showed a declining pattern (except zone 1 by 0.45 °C and zone 2 by 0.22 °C) which ranged from −0.76 °C to −0.03 °C, wherein the lowest decline zone was 2 (0.5–1 km) by −0.03 °C because there was the existence of high-density impervious land. The highest decline zone was 11 (5–5.5 km) by −0.76 °C because of the presence of forest and grassland. **Between W2 and W3**, the difference in mean LST was found to decline, which ranged from −2.24 °C to −0.48 °C, wherein the lowest decline zone was 20 (city center) by −0.48 °C because of the presence of bare soil with scattered grassland. The highest decline zone was 0 (city center) by −2.24 °C because of the high moisture content over scattered vegetation in the high-density impervious land. **Between W3 and W4**, the difference in mean LST was found amplified in the range 0.64 °C to 2.48 °C, wherein the lowest intensified zone was 17 (8–8.5 km) by 0.64 °C because of the presence of Ganga River flow. The highest intensified zone was 22 (10.5–11 km) by 2.48 °C because of the dominant barren land. **Between W1 and W4**, the difference in mean LST exhibited a very interesting amplifying pattern [except for zone 9 (4–4.5 km), zone 11 (5–5.5 km), and zone 15 to zone 19 (7–9.5 km), which ranged from −0.89 °C to −0.21 °C as these zones had forest and grassland, as well as Ganga River flow] which ranged from 0.02 °C to 1.06 °C, wherein the lowest intensified zone was 8 (3.5–4 km) by 0.02 °C because there was the existence of impervious land. The highest intensified zone again was 22 (10.5–11 km) by 1.06 °C due to the dominant barren land.

### 3.3. Spatiotemporal Dynamics of Land Indices and LST, and Their Relationships

#### 3.3.1. NDBI Dynamics and Its Connection with LST

The spatial NDBI distributional dynamics maps of Prayagraj are shown in Figure 7a,b for the summer season for S1, S2, S3, and S4 time points and the winter season for W1, W2, W3, and W4, respectively. The statistics of all the six land indices and their relationship with LST for all the summer and winter time points are presented in Table 6. Furthermore, Figure A1 shows whisker boxplots of all the land indices for the summer/winter seasons. The analysis found that summer mean NDBI witnessed a decrease of −0.023 in S1 which amplified to −0.015, −0.02, and 0.02 in S2, S3 and S4, respectively. The winter mean NDBI witnessed −0.036 in W1 which amplified to −0.012 in W2 but declined to −0.018 in W3 and further to −0.039 in W4.

**Table 6.** Statistics of the six land indices and their relationship with LST for all the summer and winter time points.

| Season | Time Points | Land Indices | Minimum | Maximum | Mean | Standard Deviation | Correlation with LST[®] | Significance (*p*) |
|---|---|---|---|---|---|---|---|---|
| Summer | S1 | NDBI | −0.324 | 0.130 | −0.023 | 0.055 | 0.668 | <0.001 |
| | | EBBI | 0.051 | 0.240 | 0.158 | 0.027 | 0.623 | <0.001 |
| | | NDMI | −0.130 | 0.324 | −0.023 | 0.055 | −0.668 | <0.001 |
| | | NDVI | −0.098 | 0.521 | 0.134 | 0.068 | −0.459 | <0.001 |
| | | NDWI | −0.463 | 0.132 | −0.168 | 0.057 | 0.285 | <0.001 |
| | | SAVI | −0.048 | 0.363 | 0.088 | 0.043 | −0.425 | <0.001 |
| | S2 | NDBI | −0.407 | 0.184 | −0.015 | 0.073 | 0.6758 | <0.001 |
| | | EBBI | 0.010 | 0.276 | 0.149 | 0.038 | 0.640 | <0.001 |
| | | NDMI | −0.184 | 0.407 | −0.015 | 0.073 | −0.6758 | <0.001 |
| | | NDVI | −0.196 | 0.661 | 0.180 | 0.093 | −0.266 | <0.001 |
| | | NDWI | −0.575 | 0.274 | −0.202 | 0.080 | 0.070 | <0.001 |
| | | SAVI | −0.070 | 0.462 | 0.111 | 0.057 | −0.259 | <0.001 |
| | S3 | NDBI | −0.366 | 0.189 | −0.020 | 0.058 | 0.6757 | <0.001 |
| | | EBBI | 0.042 | 0.263 | 0.153 | 0.033 | 0.751 | <0.001 |
| | | NDMI | −0.189 | 0.366 | −0.020 | 0.058 | −0.6757 | <0.001 |
| | | NDVI | −0.096 | 0.562 | 0.143 | 0.080 | −0.376 | <0.001 |
| | | NDWI | −0.492 | 0.136 | −0.160 | 0.070 | 0.227 | <0.001 |
| | | SAVI | −0.041 | 0.391 | 0.091 | 0.049 | −0.345 | <0.001 |
| | S4 | NDBI | −0.339 | 0.188 | 0.020 | 0.060 | 0.636 | <0.001 |
| | | EBBI | 0.043 | 0.292 | 0.154 | 0.034 | 0.751 | <0.001 |
| | | NDMI | −0.188 | 0.339 | 0.020 | 0.060 | −0.636 | <0.001 |
| | | NDVI | 0.003 | 0.538 | 0.205 | 0.077 | −0.277 | <0.001 |
| | | NDWI | −0.445 | 0.215 | −0.202 | 0.056 | 0.272 | <0.001 |
| | | SAVI | 0.002 | 0.392 | 0.138 | 0.051 | −0.215 | <0.001 |
| Winter | W1 | NDBI | −0.783 | 0.249 | −0.036 | 0.107 | 0.308 | <0.001 |
| | | EBBI | 0.000 | 0.184 | 0.074 | 0.026 | 0.520 | <0.001 |
| | | NDMI | −0.249 | 0.783 | −0.036 | 0.107 | −0.308 | <0.001 |
| | | NDVI | −0.305 | 0.683 | 0.217 | 0.117 | 0.113 | <0.001 |
| | | NDWI | −0.477 | 0.526 | −0.065 | 0.107 | −0.259 | <0.001 |
| | | SAVI | −0.072 | 0.358 | 0.089 | 0.051 | 0.191 | <0.001 |
| | W2 | NDBI | −0.814 | 0.243 | −0.012 | 0.115 | 0.467 | <0.001 |
| | | EBBI | 0.000 | 0.190 | 0.075 | 0.027 | 0.564 | <0.001 |
| | | NDMI | −0.243 | 0.814 | −0.012 | 0.115 | −0.467 | <0.001 |
| | | NDVI | −0.271 | 0.611 | 0.183 | 0.103 | −0.072 | <0.001 |
| | | NDWI | −0.512 | 0.399 | −0.159 | 0.094 | −0.074 | <0.001 |
| | | SAVI | −0.060 | 0.314 | 0.072 | 0.041 | −0.003 | <0.001 |
| | W3 | NDBI | −0.597 | 0.210 | −0.018 | 0.098 | 0.536 | <0.001 |
| | | EBBI | 0.000 | 0.203 | 0.081 | 0.029 | 0.685 | <0.001 |
| | | NDMI | −0.210 | 0.597 | −0.018 | 0.098 | −0.536 | <0.001 |
| | | NDVI | −0.165 | 0.594 | 0.134 | 0.083 | 0.159 | <0.001 |
| | | NDWI | −0.512 | 0.249 | −0.110 | 0.080 | −0.369 | <0.001 |
| | | SAVI | −0.046 | 0.322 | 0.058 | 0.037 | 0.215 | <0.001 |
| | W4 | NDBI | −0.694 | 0.352 | −0.039 | 0.108 | 0.503 | <0.001 |
| | | EBBI | 0.000 | 0.554 | 0.079 | 0.032 | 0.749 | <0.001 |
| | | NDMI | −0.352 | 0.694 | −0.039 | 0.108 | −0.503 | <0.001 |
| | | NDVI | −0.282 | 0.623 | 0.163 | 0.110 | 0.032 | <0.001 |
| | | NDWI | −0.524 | 0.367 | −0.148 | 0.102 | −0.186 | <0.001 |
| | | SAVI | −0.145 | 0.560 | 0.125 | 0.081 | 0.080 | <0.001 |

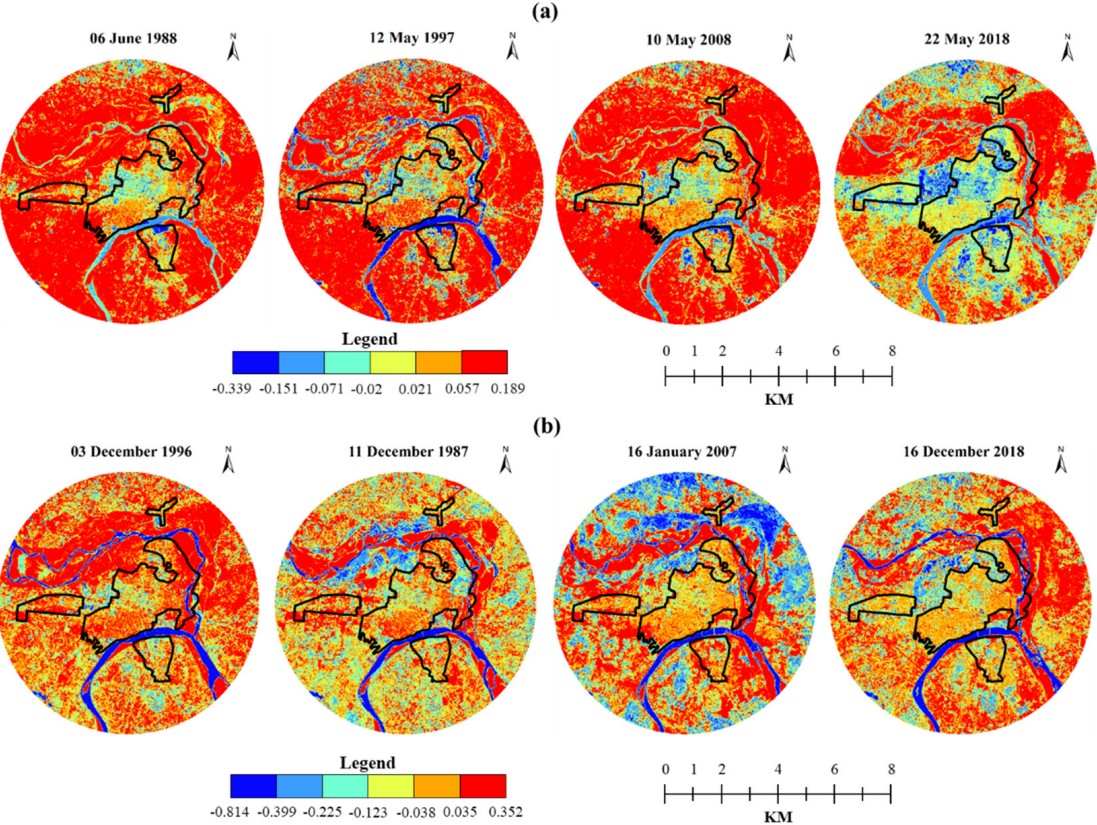

**Figure 7.** Seasonal NDBI dynamics in Prayagraj city (1987–2018): (**a**) summer and (**b**) winter.

Most of the highly dense impervious/built-up area was intensely concentrated in the city center to the south in S1 and W1, which further expanded in the south in S2, W2, S3, and W3. In S4 and W4, the highly dense built-up area spread in the southwest and northeast in S4 and W4, respectively. Therefore, it is apparent that the built-up area is mostly concentrated in the southwest and northeast up to 8 km.

The correlations between LST and all six land indices for the summer and winter seasons are presented in Figures 13 and 14, respectively. In the summer/winter seasons, the correlations between LST vs. NDBI were found to be positive at all distinctive time points. On the basis of the values of correlation coefficients, it can be found that the NDBI's role in LST intensification in the summer was higher than in the winter. Further, the built-up intensity effect was amplified over the city landscape, resulting in increased temperature growth because of the conversion of forests and water bodies into built-up land.

### 3.3.2. EBBI Dynamics and Its Connection with LST

The spatiotemporal maps of EBBI dynamics are shown in Figure 8a,b for the summer/winter seasons, respectively, for all the distinctive summer/winter time points (however, summer/winter seasonal statistics are shown in Figure A1). The summer mean EBBI witnessed an amplifying trend in the summer/winter seasons at all time points. The highly dense impervious/built-up area was mostly concentrated in the city center to 4 km of its periphery, and the bare land was mostly concentrated to 6–8 km and 9.5–11 km in both S1 and W1. Then, the highly dense impervious/ built-up area expanded in the south and north up to 6 km in S2 and W2, and bare land mostly remained concentrated in the northeast (6–8 km) and southwest (6–8 km). In S3 and W3, the highly-dense impervious/built-up area expanded in the south and north up to 7 km, and bare land mostly remained concentrated in northeast (6–8 km) and southwest (7–9 km). However, the highly dense impervious/built-up area expanded in the southwest, northeast, and northwest up to 8 km in S4 and W4, and bare land mostly remained concentrated on northeast (6–8 km) and

southwest (8–9 km). Therefore, it is apparent that the impervious/built-up area mostly existed in the southwest and northeast up to 7 km.

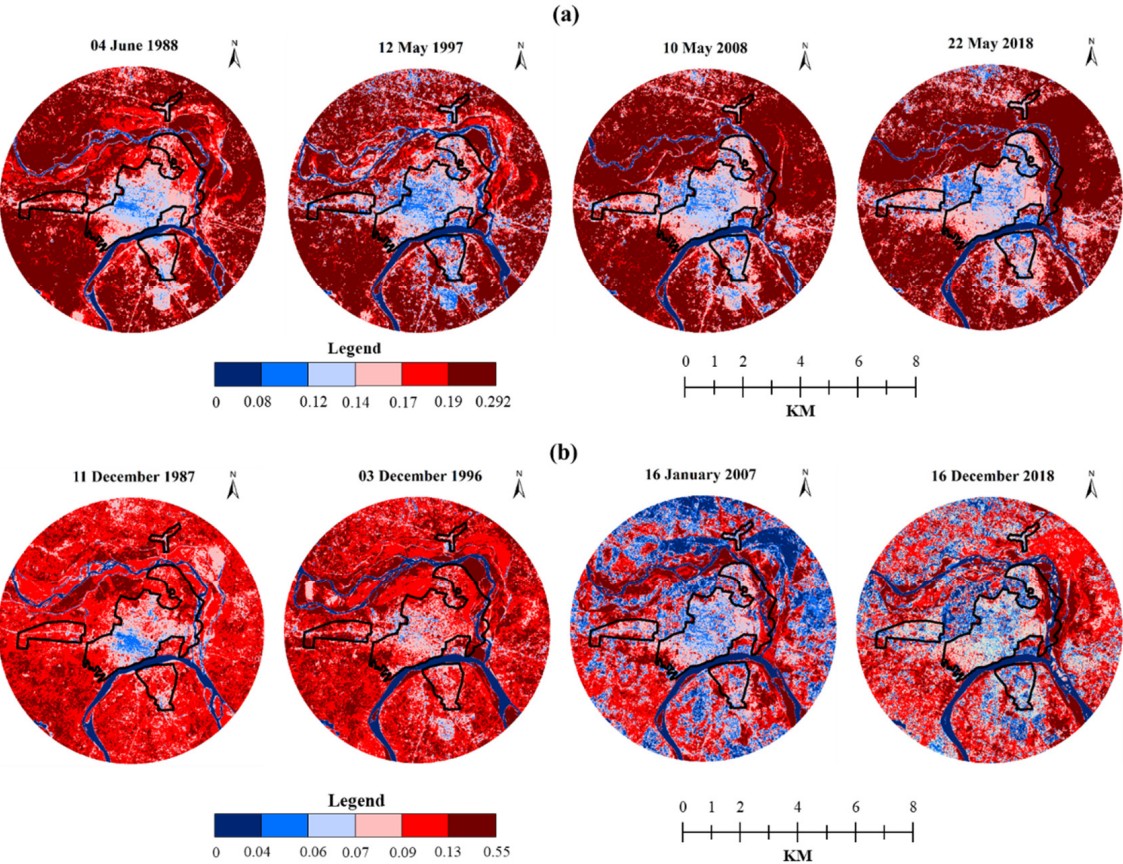

**Figure 8.** Seasonal EBBI dynamics in Prayagraj city (1987–2018): (**a**) summer and (**b**) winter.

A positive correlation was detected between LST vs. EBBI at all distinctive summer/winter time points (Figures 13 and 14). The role of EBBI in LST intensification remained high in both the summer and the winter (Table 6). This indicates that the impervious/built-up and bare land intensity effect was amplified over the city landscape, resulting in increased temperature growth due to the conversion of forests, water bodies, and agricultural land into built-up land and bare land settings.

### 3.3.3. NDMI Dynamics and Its Relationship with LST

Seasonal NDMI dynamics are shown in Figure 9a,b for the summer and winter seasons, respectively (however, summer/winter seasonal statistics are shown in Figure A1). The summer mean NDMI witnessed a nonuniform pattern where it first amplified in S1 and S2 but declined in S3, before again amplifying in S4. In contrast, the winter mean NDMI shows an increasing pattern in W1 and W2 but a decreasing one in W3 and W4.

This may be attributed to the fact that the high moisture content area was concentrated in the northwest and northeast in S1 and W1. A decrease was observed in the moisture content area in the northwest in S2 and W2 which further diminished in S3 and W3. However, the high moisture content in the northwest and southeast was again amplified in S4 and W4. In fact, the forest, grassland, and Ganga River with high moisture content were primarily present in the city landscape.

In both seasons, the correlation between LST vs. NDMI was found to be negative at all time points (Figures 13 and 14), while the correlation coefficient values (Table 6) indicate that the role of NDMI in decreasing LST was higher in the summer season than in the winter season.

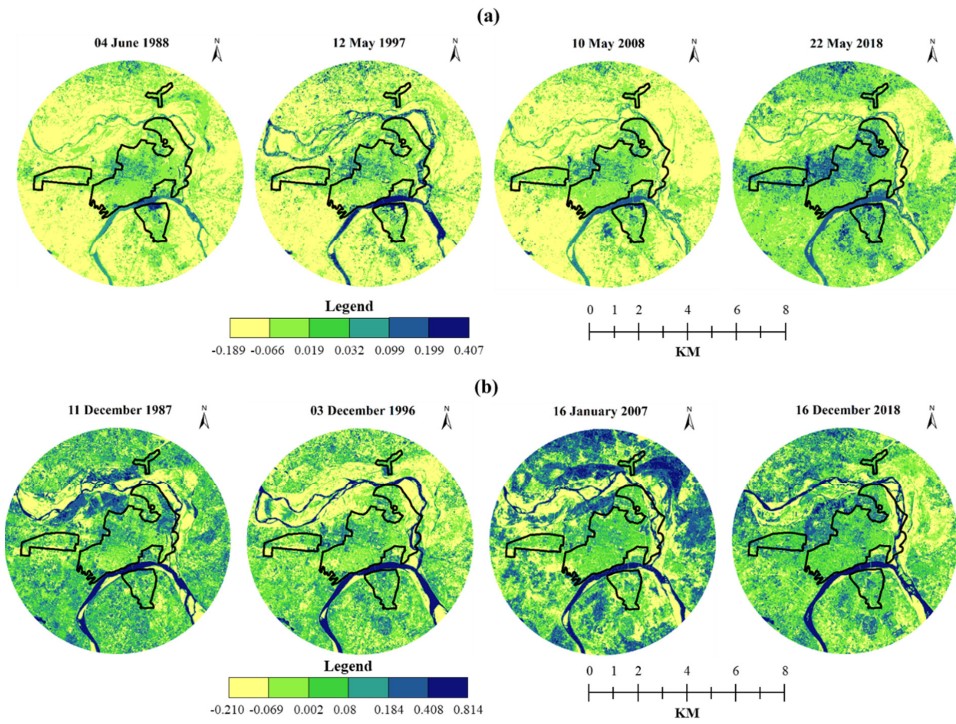

**Figure 9.** Seasonal NDMI dynamics in Prayagraj city (1987–2018): (**a**) summer and (**b**) winter.

### 3.3.4. NDVI Dynamics and Its Relationship with LST

The NDVI maps of Prayagraj are shown in Figure 10a,b for the summer and winter seasons, respectively (however, summer/winter seasonal statistics are shown in Figure A1). An increase was observed in the summer mean NDVI at S2, followed by a decrease in S3 but again an increase in S4. However, the winter mean NDVI witnessed a decline in W2 and W3 but an increase in W4.

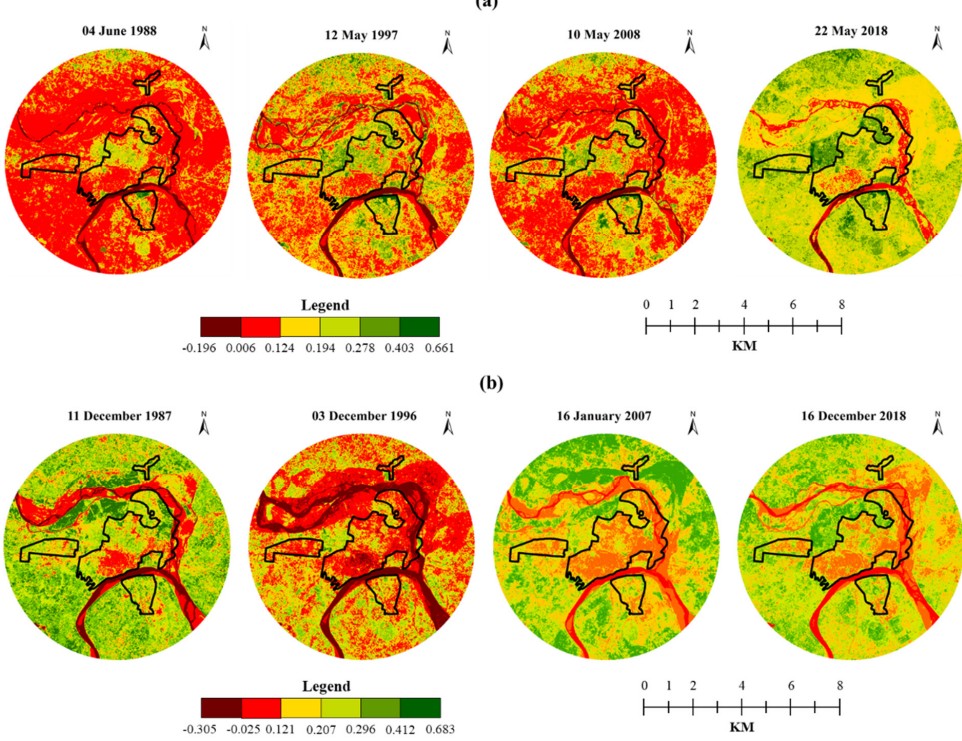

**Figure 10.** Seasonal NDVI dynamics in Prayagraj city (1987–2018): (**a**) summer and (**b**) winter.

The analysis shows that high-density vegetation was concentrated in the northwest in S1 and W1, showing a diminishing trend in S2, W2, S3, and W3 but an amplification in S4 and W4. It was found that the forest and grassland were mainly dominant in the northwest direction from the city center.

In the summer season, the correlations between LST vs. NDVI were found to be negative at all summer time points (Figure 13), while, in the winter season, the correlation between LST vs. NDVI was also positive at all winter time points except for W2 (Figure 14). This reflects that NDVI played a significant role in decreasing LST in the summer season, but its role was very weak in the winter.

### 3.3.5. NDWI Dynamics and Its Relationship with LST

The NDWI maps of Prayagraj are shown in Figure 11a,b for the summer and winter seasons for all time points, respectively (however, summer/winter seasonal statistics are shown in Figure A1). The summer and winter mean NDWI followed a similar pattern, first declining in S2 and W2, then amplifying in S3 and W3 before again declining in S4 and W4.

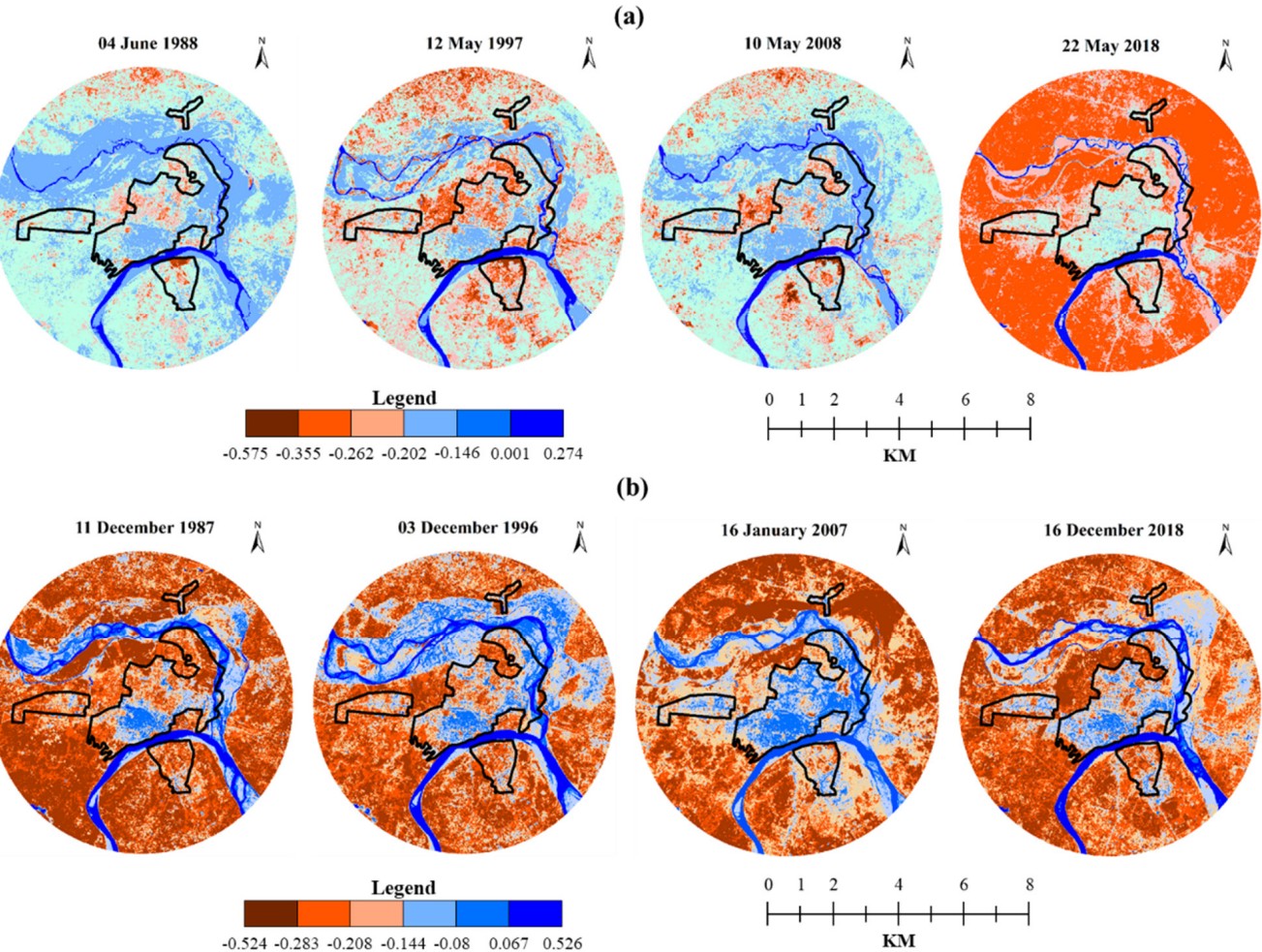

**Figure 11.** Seasonal NDWI dynamics in Prayagraj city (1987–2018): (**a**) summer and (**b**) winter.

High-density water bodies were concentrated in the northeast (8–9.5 km) due to the Ganga River flow in S1 and W1. The high-density vegetation was concentrated in the northwest and southeast, which decreased in S2, W2, S3, and W3. Then, high-density vegetation increased in the northwest and southeast in S4 and W4, respectively. Therefore, it is apparent that the forest and grassland were mainly dominant in the northwest direction from the city center.

In both summer and winter seasons, the correlations between LST vs. NDWI was positive at all times except in W2 but with relatively low correlation coefficient values (Figures 13 and 14). This means that NDWI played an insignificant role in decreasing LST in both summer and winter because the whole city exhibited a lack of water bodies except for 8–9.5 km in the northeast direction of the Ganga River flow.

### 3.3.6. SAVI Dynamics and Its Relationship with LST

The distributional dynamics maps of SAVI are shown in Figure 12a,b for the summer and winter seasons, respectively (however, summer/winter seasonal statistics are shown in Figure A1). The summer mean SAVI amplified in S2 but declined in S3 before again amplifying in S4. The winter mean SAVI declined in W2 and W3 but amplified in W4.

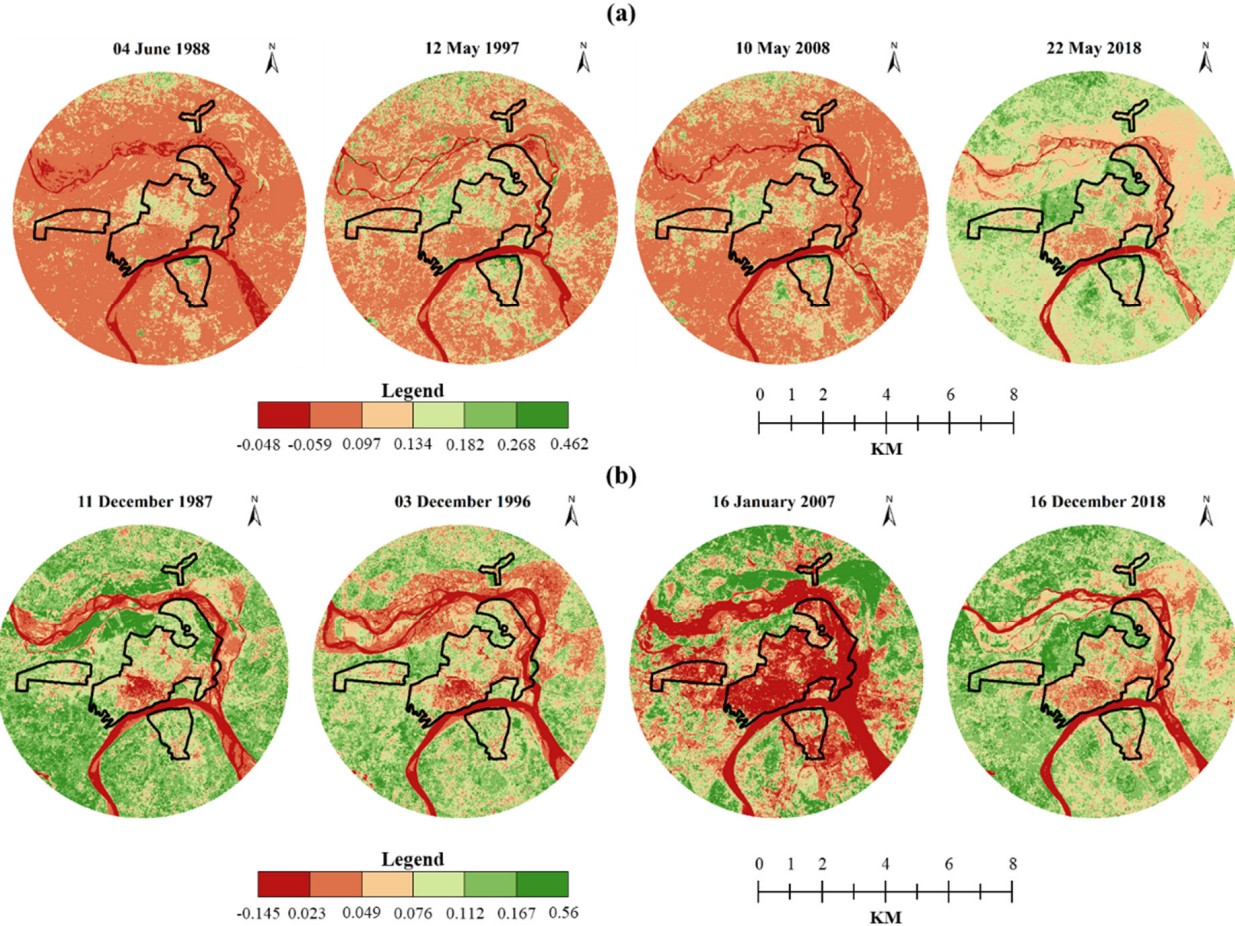

**Figure 12.** Seasonal SAVI Dynamics in Prayagraj city (1987–2018): (**a**) summer and (**b**) winter.

It can be observed that high-density forest and grassland were concentrated in the northwest and southeast in S1 and W1, which was reduced in S2 and W2. The high-density forest and grassland concentrated in the northwest diminished in S3 and W3. However, the high-density forest and grassland concentrated within the northwest and southeast increased in S4 and W4. Therefore, it was observed that the forest and grassland mostly dominated in the northwest direction from the city center.

In the summer season, the correlation between LST vs. SAVI was found to be negative at each distinctive summer time point (Figure 13), while, in the winter season, the correlation between LST vs. SAVI was found to be positive at all winter time points except in W2 (Figure 14). This means that SAVI played a noteworthy role in decreasing LST in the summer season, but its role was very weak in the winter.

## 3.4. Effects of Land Indices on LST Distribution

The effect of spatiotemporal seasonal dynamics of all six land indices (i.e., NDBI, EBBI, NDMI, NDVI, NDWI, and SAVI) on LST profiling for summer/winter seasons from the city center to the periphery in eight different directions, i.e., north to south, northeast to southwest, northwest to southeast, and west to east was extracted on the city landscape. The effect of these six land indices on the LST in the summer and winter seasons is presented in Figures 15 and 16, respectively.

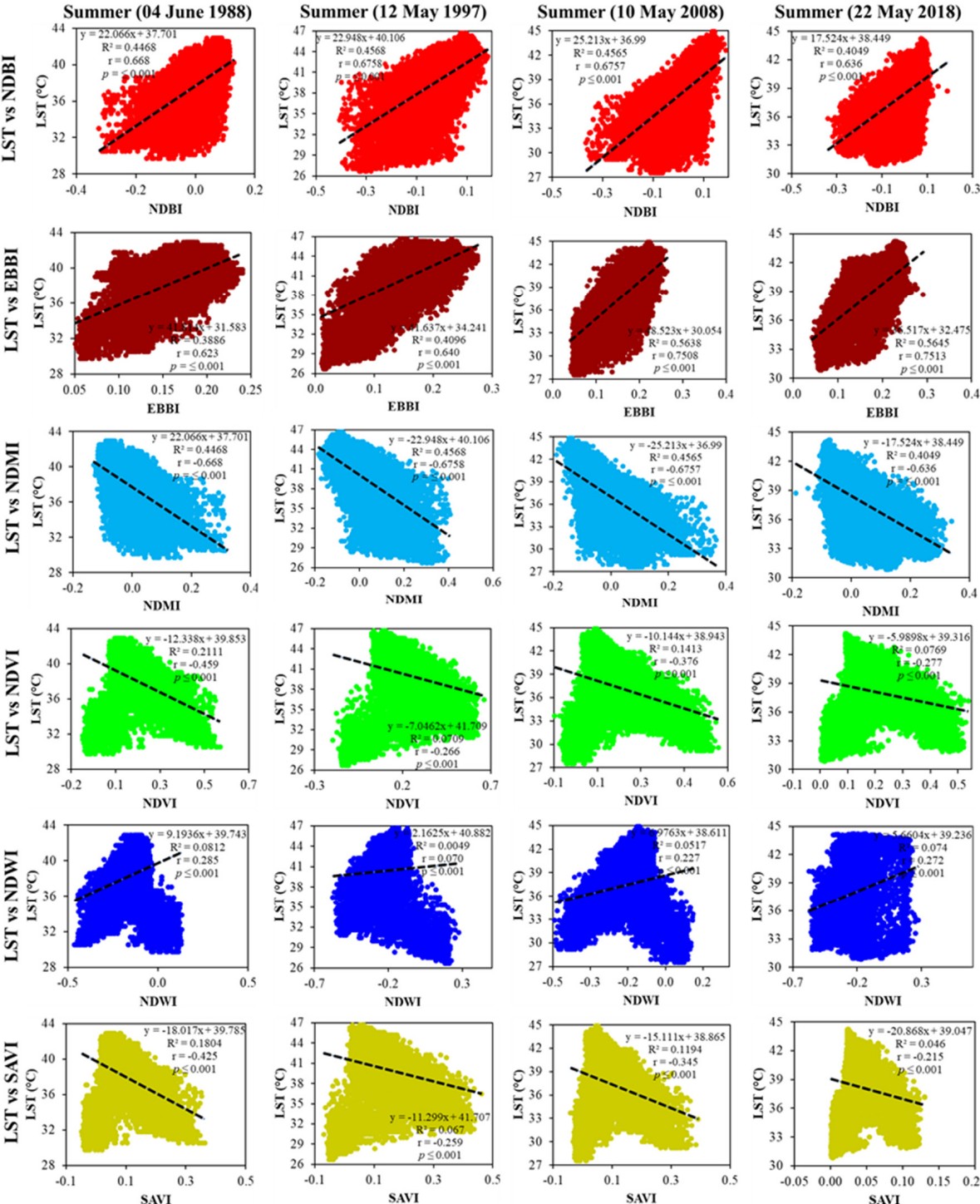

**Figure 13.** Correlation between the LST and the six land indices of Prayagraj city in the summer season (1988–2018).

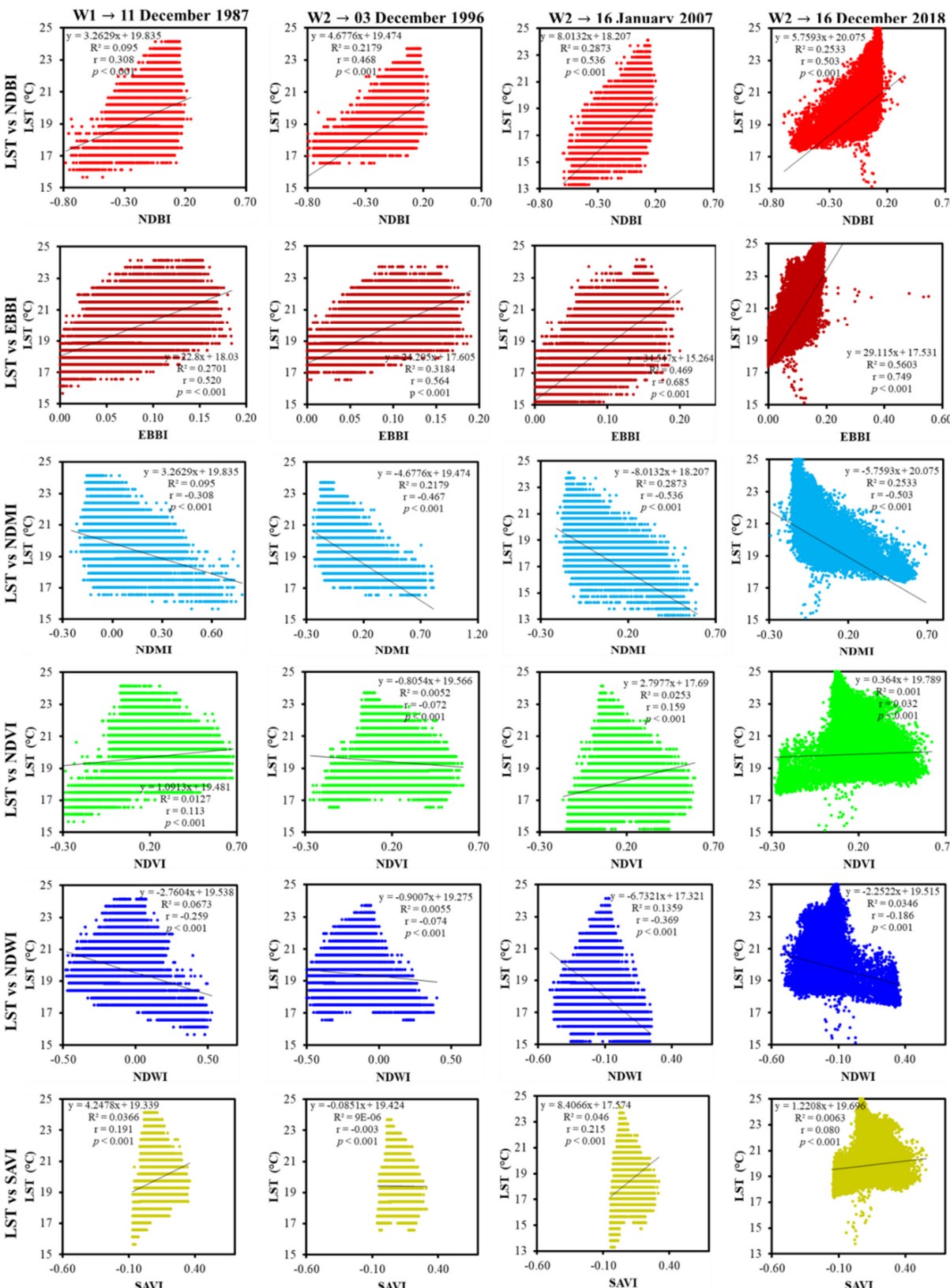

**Figure 14.** Correlation between the LST and the six land indices of Prayagraj city in the winter season (1987–2018).

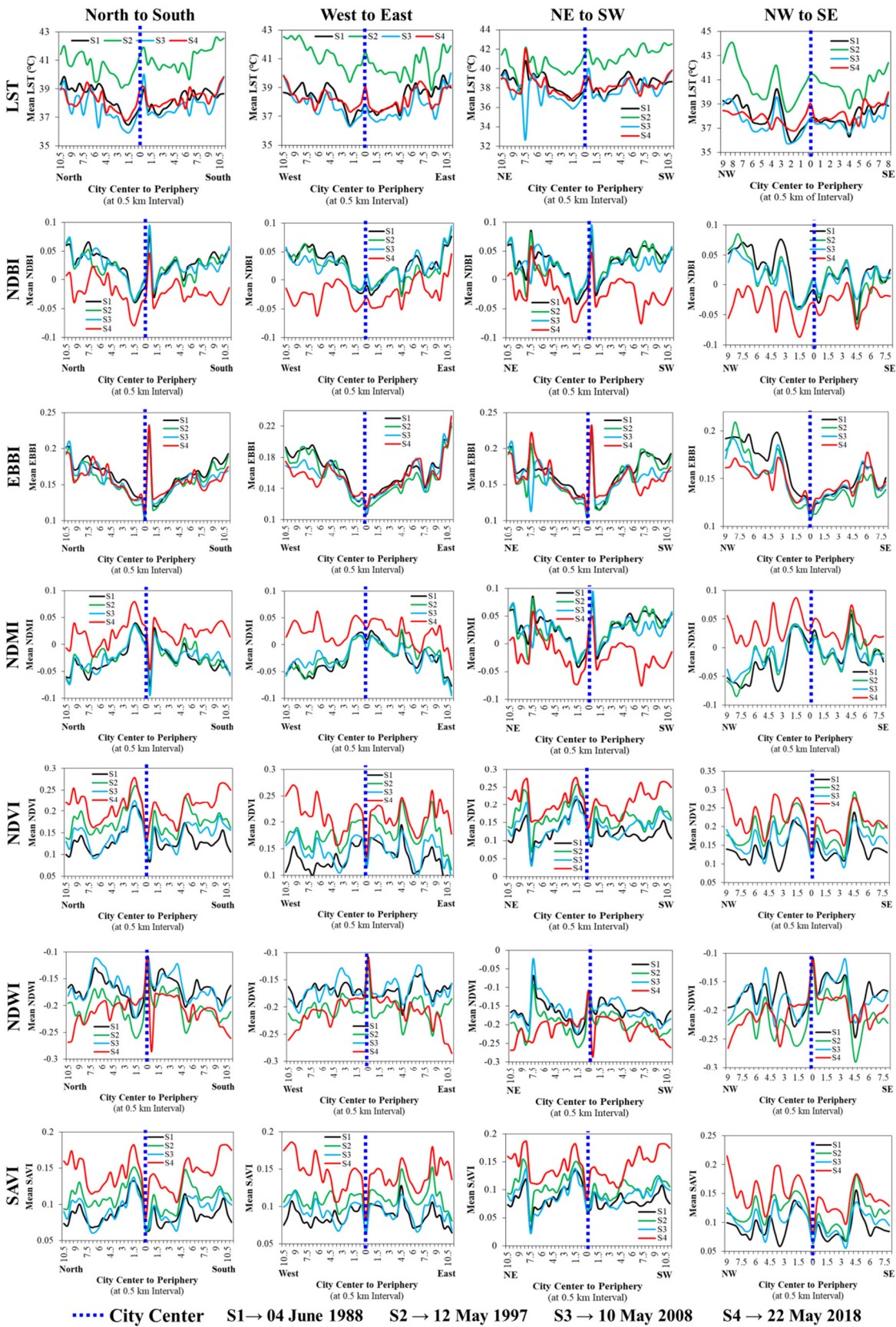

**Figure 15.** Effects of the land indices on the distribution of LST dynamics in the summer season in Prayagraj city (1988–2018).

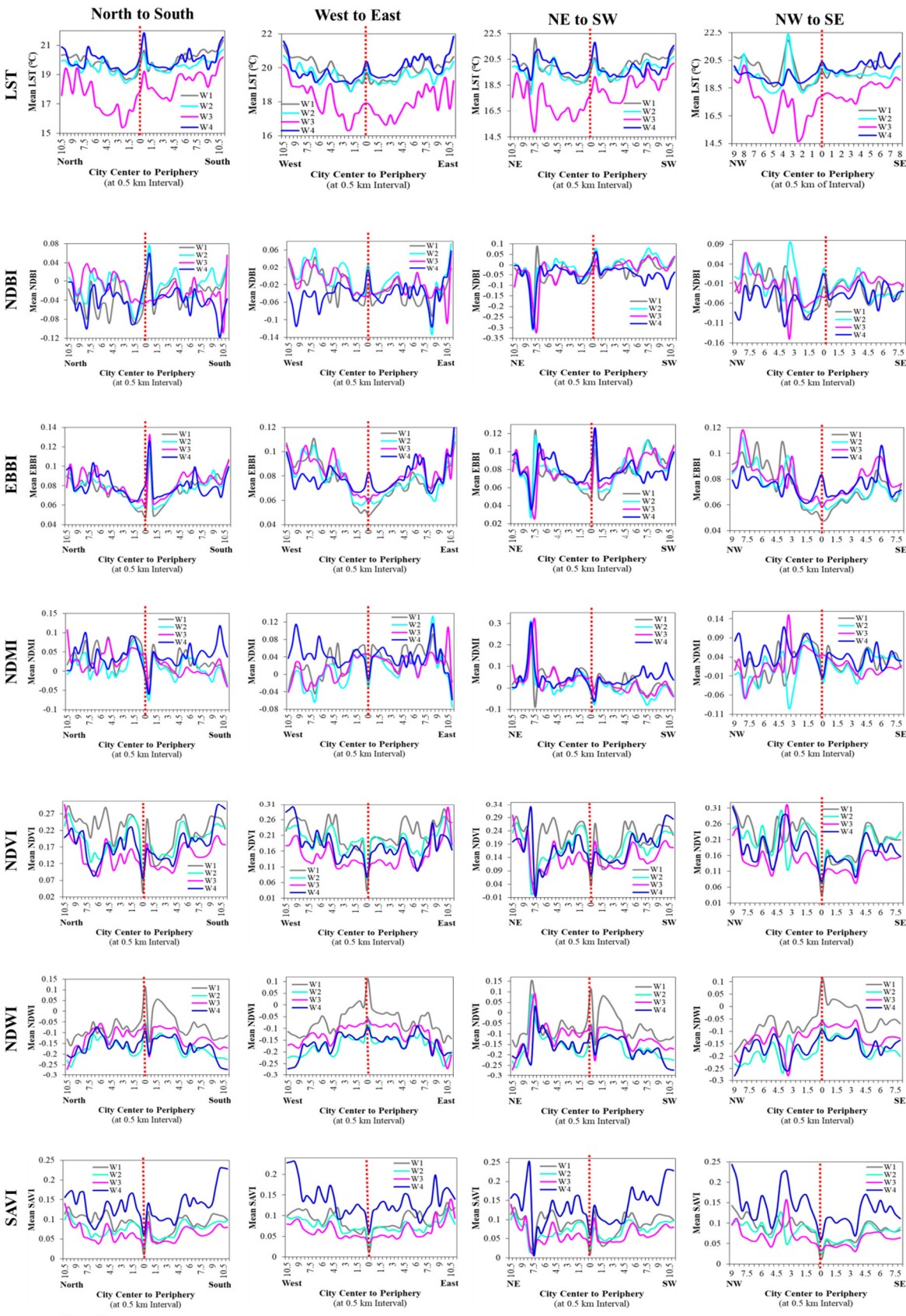

**Figure 16.** Effects of the land indices on the distribution of LST dynamics in the winter season in Prayagraj city (1987–2018).

3.4.1. North to South

**In summer**, from the city center in both the north and the south directions, the mean LST followed a declining trend (up to 11 km) of 0.2 °C–2.75 °C except for 6.5–11 km in the north direction and 9–11 km in the south direction (Figure 15). The higher mean LST at the city center was because of the impervious/built-up land (high mean NDBI by −0.04 to 0.04). The decline in mean LST witnessed at 1–2 km in the north and 4.5–6.5 km in the south was due to the existence of forest land/grassland (higher mean NDVI by 0.02–0.10), as well as the availability of higher moisture content (higher mean NDMI and mean SAVI). However, the increase in mean LST at 6.5–7.5 km and 10–11 km in the north and 6.5–11 km in the south was because of the existence of barren land/bare soils (higher mean EBBI by 0.01–0.08). The mean NDWI was found to be negative ($<-0.1$) at all consecutive summer time points, which means that its role was insignificant due to the unavailability of water bodies of large size (except rivers) to impact the reduction in mean LST.

**In winter**, from the city center in both the north and the south directions, the mean LST witnessed declining inclination up to 11 km by 0.2 °C–3.6 °C except for 6.5–9.5 km in the north, and 5–9 km and 10–11 km in the south (Figure 16). The higher mean LST in the city center was due to the built-up land presence (high mean NDBI by −0.04 to 0.04). In contrast, the decline in mean LST witnessed at 1–2 km and 9.5–11 km in the north and 9–10 km in the south was because of the presence existence of grassland/forest land (higher mean NDVI by 0.02–0.25) coupled with high moisture content availability, as reflected by high mean NDMI and mean SAVI values. The existence of barren land/bare soils (higher mean EBBI by 0.01–0.08) at 6.5–9.5 km in the north and at 5–9 km and 10–11 km in the south resulted in a spike in mean LST. However, as in the summer season, mean NDWI had negative values in the winter season also [<0 in the north and south (except at city center and 1.5–2 km in the south)] at all consecutive winter time points, signifying its insignificant role in the reduction in mean LST.

3.4.2. Northeast (NE) to Southwest (SW)

**In summer**, in both the NE and the SW directions from the city center, mean LST detected a declining pattern up to 11 km by 0.2 °C–5.8 °C except for 6–7.5 km in the NE and 6.5–11 km in the SW, where a reverse trend was witnessed. The decline in mean LST witnessed at 1–3 km in the NE and 6.5–7.5 km and 9.5–10.5 km in the SW was because of the availability of grassland/forest land (higher mean NDVI by 0.02–0.13) and high moisture, as the mean NDMI and mean SAVI also witnessed a similar pattern to mean NDVI. The availability of barren land/bare soils (higher mean EBBI by 0.01–0.1) resulted in the increase in mean LST at 6.5–7.5 km in the NE and 7.5–9.5 km in the SW. The mean NDWI showed negative values in the summer, indicating that its role was insignificant due to the unavailability of water bodies. However, 7.5–8 km in the NE showed the reverse trend due to the Ganga River flow.

**In winter**, in both the NE and the SW directions from the city center, the mean LST witnessed a declining pattern up to 11 km by 0.2 °C–4.5 °C except for 6–7.5 km in the NE direction and 10–11 km in the SW direction where the mean LST witnessed an increasing trend (Figure 16). The city center had high-density built-up land (higher mean NDBI by −0.04 to 0.04), leading to higher mean LST which witnessed a decline at 1–3 km in the NE and 4.5–6 km and 10–11 km in the SW because of the existence of grassland/forest land (higher mean NDVI by 0.02–0.2). The mean NDMI and mean SAVI also witnessed a similar pattern to that of mean NDVI, indicating the availability of higher moisture due to the presence of forest/grassland, which helped in the decline in mean LST. However, the mean LST high spikes at 6–7.5 km in the NE direction and at 10–11 km in the SW direction were due to barren land/bare soils (higher mean EBBI by 0.01–0.08) at these locations. However, in summer, the role of NDWI was insignificant except for 7.5–8.5 km in the NE due to Ganga River.

### 3.4.3. Northwest (NW) to Southeast (SE)

**In summer**, in both the NW and the SE directions from the city center, it was detected that the mean LST witnessed a declining pattern up to 9 km in the NW and 8 km in the SE by 0.1 °C–2.6 °C because of the existence of grassland/forest land (higher mean NDVI by 0.01–0.15) coupled with high moisture availability due to high mean NDMI and mean SAVI (Figure 15). However, an increasing pattern was witnessed in the zones of 3.5–4.5 km and 6–9 km in the NW and 6.5–8 km in the SE because of the presence of barren land/bare soils (higher mean EBBI by 0.01–0.2). The role of the mean NDWI was insignificant due to negative values (<0.1) at all consecutive summer time points.

**In winter**, in both the NW and the SE directions from the city center, the mean LST witnessed a declining pattern up to 11 km by 0.1 °C–3.5 °C except for 3–5 km and 6.5–9 in the NW direction and 5–8 km in the SE direction, where the mean LST witnessed an increasing pattern. The decline in mean LST at 1–3 km in the NW and 4.5–6.5 km and 3–5 km in the SE was because of the presence of grassland/forest land (higher mean NDVI by 0.02–0.23) and the high peaks of mean NDMI and mean SAVI (Figure 16). The barren land/bare soils resulted in an increase in mean LST inclination at 3–5 km and 6.5–9 km in the NE and at 5–8 km in the SE (higher mean EBBI by 0.01–0.06). Again, the role of NDWI was insignificant due to negative values of the mean NDWI [except the city center to 1.5 km in the SE at W1, where the mean LST was >0] at all consecutive time points.

### 3.4.4. West to East

**In summer**, the mean LST witnessed a higher peak at the city center in comparison to nearby surroundings because of the presence incidence of high-density impervious/built-up land (higher mean NDBI by −0.04 to 0.04). Then, the mean LST followed a declining pattern up to 11 km by 0.1 °C–1.75 °C in both the west (in particular, at 1–3 km and 5.5–6.5 km) and the east (in particular, 3–5 km and 8–9.5 km) due to forest land/grassland (higher mean NDVI by 0.02–0.10) coupled with high moisture availability (Figure 15). The mean LST witnessed an increasing pattern in 4–5.5 km and 6.5–11 km in the west and 6–7.5 and 9.5–11 km in the east because of barren land/bare soils (higher mean EBBI by 0.01–0.2). However, mean NDWI was found to have negative values (<–0.1) at all consecutive summer time points, which means that its role was insignificant due to the unavailability of many water bodies.

**In winter**, the city center again witnessed higher LST compared to its surroundings. However, the mean LST witnessed declining inclination up to 11 km by 0.1 °C–1.6 °C [except 4–5.5 km and 6.5–11 km in the west and 5.5–7.5 km and 9.5–11 km in the east] in both the west and the east directions. The decline in mean LST witnessed at 1.5–3 km and 5.5–6.5 km in the west and 4–6 km and 8.5–9.5 km in the east was because of the existence of grassland/forest land (higher mean NDVI by 0.02–0.22). The mean NDMI and mean SAVI also witnessed a similar pattern to the mean NDVI, which means that higher moisture was available where forest/grassland was present, consequently helping the decline in the mean LST (Figure 16). However, the mean LST inclination at 4.5–5 km and 6.5–11 km in the west direction and at 5.5–7.5 km and 9.5–11 km in the east direction was because of barren land/bare soils presence (higher mean EBBI by 0.01–0.05). However, the mean NDWI was found to have negative values [<0 in the north and south (except at city center to 0.5 km in the west)] at all consecutive winter time points, which means that its role was insignificant due to the unavailability of water bodies large in size, which may have impacted the reduction in the mean LST.

### 3.5. SUHI Dynamics

### 3.5.1. Urban and Rural/Suburban Point Location-Based SUHI

The distribution of SUHI dynamics was evaluated in the summer season at all four time points (S1, S2, S3, and S4), as well as in the winter season at all four time points (W1, W2, W3, and W4) over Prayagraj city. For this purpose, five rural/suburban point locations, well distributed over the study area (shown in Figures 3 and 4) for sum-

mer/winter seasons, were selected to study the actual variation in LST between urban and rural/suburban areas. The seasonal SUHI dynamics for all time points of the summer and winter seasons at five rural/suburban locations in Prayagraj city are shown in Figure 17. Furthermore, the statistics of point location-based SUHI are presented in Table 7. It can be observed that these five case points showed strong SUHI evidence for the summer and the winter.

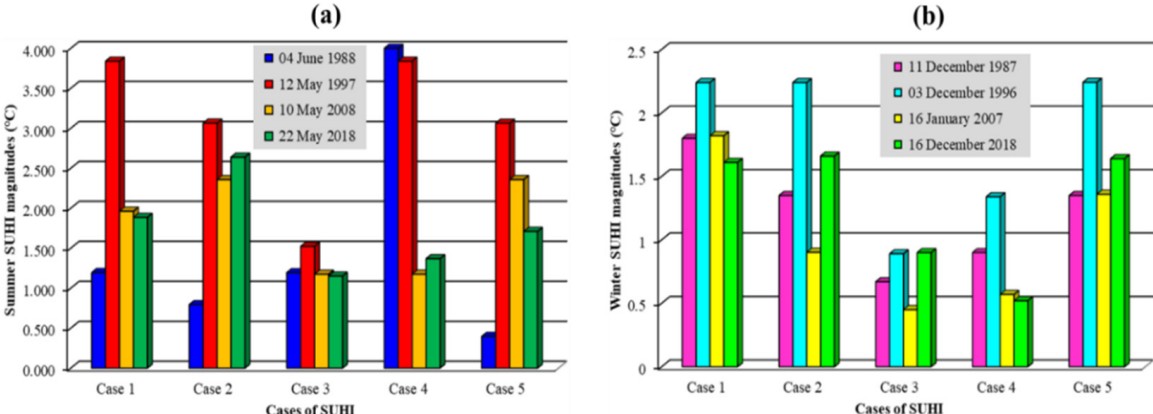

**Figure 17.** Seasonal SUHI dynamics at five case point locations in Prayagraj city (1987–2018) for all time points: (**a**) summer and (**b**) winter.

**Table 7.** Seasonal SUHI magnitude based on rural/suburban point locations in Prayagraj city.

| Season | LST (°C) Difference [$T_{U-R}$] | | | |
|---|---|---|---|---|
| **Summer SUHI** | **S1** | **S2** | **S3** | **S4** |
| **Case Point 1** | 1.195 | 3.840 | 1.962 | 1.887 |
| **Case Point 2** | 0.794 | 3.064 | 2.358 | 2.639 |
| **Case Point 3** | 1.194 | 1.524 | 1.175 | 1.151 |
| **Case Point 4** | 4.016 | 3.840 | 1.174 | 1.368 |
| **Case Point 5** | 0.398 | 3.064 | 2.358 | 1.709 |
| **Winter SUHI** | **W1** | **W2** | **W3** | **W4** |
| **Case Point 1** | 1.80 | 2.24 | 1.82 | 1.61 |
| **Case Point 2** | 1.35 | 2.24 | 0.90 | 1.66 |
| **Case Point 3** | 0.67 | 0.89 | 0.45 | 0.90 |
| **Case Point 4** | 0.90 | 1.34 | 0.57 | 0.52 |
| **Case Point 5** | 1.35 | 2.24 | 1.36 | 1.64 |

3.5.2. Directional Ring Profiling of LST for Investigation of SUHI

In the present work, directional ring profiling was used to delineate the actual difference of LST from the city center to its periphery by taking eight-directional ring profiling, i.e., north to south, NE to SE, NW to SE, and west to east. The distribution of LST dynamics was evaluated to investigate the SUHI state in the summer and winter at all four time points over Prayagraj city. Figure A2 shows the seasonal LST profiling for SUHI formation in Prayagraj city (1987–2018) for both the summer and the winter seasons.

North to South

In summer, *at S1*, the mean LST was higher in the city center than in both the north and the south directions by ~0.75–1.5 °C [except for 5–11 km in the north and 6.5–11 km in the south because these areas were constituted mostly of sands and bare soil]. *At S2*, the

mean LST was higher in the city center than in both the north and the south directions by ~0.2–2.75 °C [except for 9.5–10.5 km in the north and 7–11 km in the south because these areas were mostly sands and bare soils]. *At S3*, the mean LST was higher at the city center than in both the north and the south directions by ~0.2–2 °C [except for 6.5–7 km, 8–8.5 km, and 9.5–10.5 km in the north, and 10–11 km in the south because these areas were mostly sands and bare soils]. *At S4*, the mean LST was higher at the city center than in both the north and the south directions by ~0.2–2.2 °C except for 6.5–7.5 km and 9.5–10.5 km in the north, and 6.5 km and 10–11 km in the south because these areas were mostly sands and bare soils.

In winter, *at W1*, the mean LST was higher in the city center than in both the north and the south directions by ~0.2–1.7 °C [except for 7.5–11 km in the north, and 6.5–11 km in the south because these areas were mostly sands and bare soil]. *At W2*, the mean LST was higher at the city center than in both the north and the south directions by ~0.2–1.5 °C [except at 6 km in the north and 10–11 km in the south because these areas were mostly sands and bare soils]. *At W3*, the mean LST was higher in the city center than in both the north and the south directions by ~0.2–3.6 °C [except for 8–8.5 km and 9.5–10.5 km in the north, and 10–11 km in the south because these areas were covered mostly with sands and bare soils]. *At W4*, the mean LST was higher in the city center than in both the north and the south directions by ~0.3–2.1 °C except for 10.5–11 km in the south because of the existence of barren land/bare soils.

NE to SE

In summer, the mean LST showed a higher peak at the city center than in both the NE and the SW directions at all four time points, i.e., S1, S2, S3, and S4, by ~0.1–1.2 °C, ~0.1–4.1 °C, ~0.12–5.8 °C, and ~0.2–1.8, respectively. However, because of the existence of sand, barren land, and bare soil, a reverse pattern was observed in zones 5–7.5 km and 9–11 km in the NE, and 5–11 km in the SW at the S1 timepoint, in zones 6–6.5 km and 10–11 km in the NE, and 5.5 km and 6.5–11 km in the SW at the S2 timepoint, in zones 7 km and 9.5–10.5 km in the NE and 10–11 km in the SW at the S3 timepoint, and in zones 6.5–8 km in the NE and 6.5 km and 10–11 km in the SW at the S4 timepoint.

In winter, the mean LST in the city center also followed the same pattern as that of the summer, with higher temperatures observed as compared to suburban areas in both the NE and the SW directions. The mean LST in the city center was higher by ~0.1–3.25 °C, ~0.1–2.4 °C, ~0.2–4.5 °C, and ~0.3–2.1 °C at the W1, W2, W3, and W4 time points. However, in the NE direction, the zones 7–7.5 km at W1, 5.0 and 7.5 km at W2, 6.5–7 km, 8.5 km, and 10 km at W3, and 6–7 km at W4 exhibited a higher LST peak as compared to the city center due to the presence of sand and bare soil. Along the same line, in the SW direction, zones 4.5–11 at W1, 7–8.5 km and 9.5–11 km at W2, 4.5–5.5 km, 8 km, and 9.5–11 km at W3, and 10–11 km at W4 exhibited the same trend.

NW to SE

In summer, *at S1*, the mean LST was higher at the city center than in both the NW and the SE directions by ~0.15–1.8 °C except for zones having predominantly bare soil at 3.5–5 km and 6.5–9 km in the NW and 4.5–8 in the SE. The same pattern was also observed at the S2, S3, and S4 time points. The mean LST at the city center was higher by ~0.1–2.6 °C, ~0.2–1.3 °C, and ~0.3–2 °C in both directions at S2, S3, and S4, respectively. The zones with a reverse trend, due to the presence of predominantly bare soil, were 3.5–4.5 km and 7.5–9 km in the NW and 6–8 km in the SE at S2, 3.5–4.5 km and 7.5–9 km in the NW and 5 km and 6–8 km in the SE at S3, and 3.5–4.5 km and 7.5–9 km in the NW and 5.5–8 km in the SE at S4.

In winter, the city center observed a higher LST peak than the suburban areas in both the NW and the SE directions, as observed in the summer season. The mean LST in the city center was higher by ~0.1–1 °C, ~0.1–1.9 °C, ~0.1–3.5 °C, and ~0.3–1.7 °C at the W1, W2, W3, and W4 time points, respectively. However, exceptions to this trend were also

observed in certain zones in both the NW and the SE directions at each distinct timepoint, as can be clearly witnessed in Figure A2, because these areas were predominantly exposed to bare soil.

West to East

In summer, the city center observed higher LST than suburban areas by ~0.1–1.2 °C at S1, ~0.1–1.75 °C at S2, ~0.1–1.7 °C at S3 and ~0.1–1.5 °C at S4 in both the west and the east directions. However, because of the existence of mostly sand, barren land, and bare soil, a few zones observed a reverse trend, namely, 3.5–11 km in the west and 5–11 km in the east at S1, 4.5–5 km and 7–11 km in the west, and 7.5–8 km and 9.5–11 km in the east at S2, 4.5–5.5 km and 7–11 km in the west, and 6.5–7 km and 9.5–11 km in the east at S3, and 10–11 km in the west, and 7–8 km and 10.5–11 km in the east at S4.

In winter, the mean LST in the city center also followed the same pattern as that of the summer, with higher temperatures observed as compared to suburban areas in both the west and the east directions. The mean LST in the city center was higher by ~0.1–0.85 °C, ~0.1–1.6 °C, ~0.1–1.5 °C, and ~0.1–1.3 °C at the W1, W2, W3, and W4 time points. However, exceptions to this trend were also observed in certain zones in both the west and the east directions at each distinctive timepoint, as can be clearly seen in Figure A2, as these areas mostly had sand and bare soil.

*3.6. Hotspot Identification*

The hotspot analysis was performed using the Getis–Ord $G_i^*$ approach to analyze the spatial distribution of LST over Prayagraj city. This approach uses LST values of neighboring features and delineates both hot and cold spots over the city landscape. Hotspots are the clusters of features of high values of LST, while cold spots aggregate the features of low LST values. On the basis of this analysis, the city landscape was categorized into seven classes: very cold, cold, cool, not significant, warm, hot, and very hot.

The summer spatiotemporal hotspot maps of Prayagraj city are shown in Figure 18a for summer time points S1, S2, S3, and S4 to present the clustering distribution of hotspot dynamics, and their statistics are compiled in Table 8. The very cold spot class experienced a high loss of 1.29 km$^2$ of areal coverage during S1–S4. The cold spot class experienced a loss of 0.07 km$^2$ of areal coverage during S1–S4. The cool spot class also experienced a loss of 0.44 km$^2$ of areal coverage during S1–S4. However, the not significant class experienced an enormous gain of 7.56 km$^2$ of areal coverage during S1–S4. The warm spot class experienced a loss of 1.81 km$^2$ of areal coverage during S1–S4. The hot spot class experienced a loss of 2.06 km$^2$ of areal coverage during S1–S4. The very hot spot class experienced a loss of 1.88 km$^2$ of areal coverage during S1–S4. This summer hotspot pattern indicates that the comfort level of living space intensively decreased in the city landscape as the areas of very cold, cold, and cool spots severely declined.

The winter spatiotemporal hotspot maps of Prayagraj city are shown in Figure 18b for the W1, W2, W3, and W4 winter time points to present the clustering distribution of hotspot dynamics. Their statistics are shown in Table 8. The very cold spot class experienced a severe loss of 6.49 km$^2$ of areal coverage during S1–S4. The cold spot class experienced a loss of 0.80 km$^2$ of areal coverage during S1–S4. The cool spot class also experienced a loss of 2.08 km$^2$ of areal coverage during S1–S4. However, the not significant class has experienced a considerable gain of 15 km$^2$ of areal coverage during S1–S4. The warm spot class experienced a loss of 1.46 km$^2$ of areal coverage during S1–S4. The hot spot class experienced a loss of 1.30 km$^2$ of areal coverage during S1–S4. The very hot spot class experienced a loss of 2.87 km$^2$ of areal coverage during S1–S4. This winter hotspot pattern also indicates that the comfort level of living space was on the decline in the city landscape as the areas of very cold, cold, and cool spots severely declined.

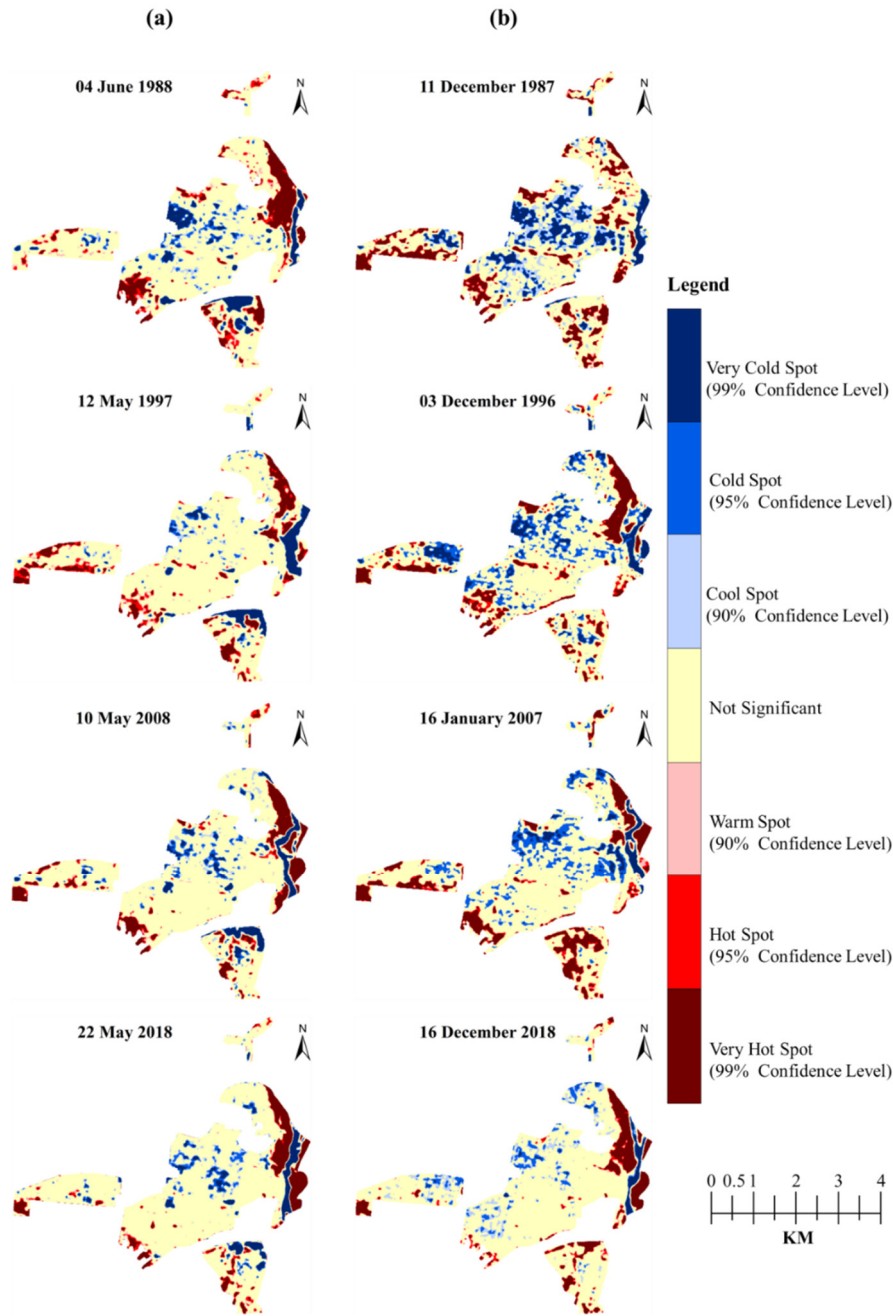

**Figure 18.** Seasonal hotspot maps based on Getis–Ord G$_i$* in Prayagraj city (1987–2018): (**a**) summer and (**b**) winter.

**Table 8.** Seasonal hotspot classes based on Getis–Ord G$_i$* statistics.

| Hot-Spot Classes Based on Getis–Ord Gi* Analysis | Area (km²) [Area (%)] | | | | |
|---|---|---|---|---|---|
| *Summer Time Points* | **S1** | **S2** | **S3** | **S4** | **Change during S1–S4** |
| Very cold spot (99% of confidence level) | 5.51 (7.55%) | 4.76 (6.52%) | 4.71 (6.45%) | 4.22 (5.78%) | −1.29 (−1.77%) |
| Cold spot (95% of confidence level) | 3.05 (4.18%) | 2.03 (2.78%) | 3.08 (4.22%) | 2.98 (4.08%) | −0.07 (−0.10%) |
| Cool spot (90% of confidence level) | 2.95 (4.04%) | 2.21 (3.03%) | 2.86 (3.92%) | 2.52 (3.45%) | −0.44 (−0.60%) |



**Table 8.** *Cont.*

| Hot-Spot Classes Based on Getis–Ord Gi* Analysis | Area (km²) [Area (%)] | | | | |
|---|---|---|---|---|---|
| Not significant | 45.35 (62.14%) | 50.67 (69.43%) | 49.98 (68.48%) | 52.90 (72.49%) | 7.56 (10.36%) |
| Warm spot (90% of confidence level) | 3.17 (4.34%) | 2.62 (3.59%) | 1.63 (2.23%) | 1.36 (1.86%) | −1.81 (−2.48%) |
| Hot spot (95% of confidence level) | 4.00 (5.48%) | 3.96 (5.43%) | 2.65 (3.63%) | 1.94 (2.66%) | −2.06 (−2.82%) |
| Very hot spot (99% of confidence level) | 8.94 (12.25%) | 6.73 (9.22%) | 8.07 (11.06%) | 7.06 (9.67%) | −1.88 (−2.58%) |
| *Winter Time Points* | **W1** | **W2** | **W3** | **W4** | **Change during W1–W4** |
| Very cold spot (99% of confidence level) | 7.97 (10.92%) | 5.40 (7.40%) | 3.35 (4.59%) | 1.49 (2.04%) | −6.49 (−8.89%) |
| Cold spot (95% of confidence level) | 4.16 (5.70%) | 7.83 (10.73%) | 7.67 (10.51%) | 3.36 (4.60%) | −0.80 (−1.10%) |
| Cool spot (90% of confidence level) | 7.44 (10.19%) | 4.31 (5.91%) | 2.94 (4.03%) | 5.36 (7.34%) | −2.08 (−2.85%) |
| Not significant | 36.32 (49.77%) | 41.30 (56.59%) | 43.62 (59.77%) | 51.32 (70.32%) | 15.00 (20.55%) |
| Warm spot (90% of confidence level) | 2.83 (3.88%) | 0.83 (1.14%) | 1.81 (2.48%) | 1.37 (1.88%) | −1.46 (−2.00%) |
| Hot spot (95% of confidence level) | 3.42 (4.69%) | 3.88 (5.32%) | 3.22 (4.41%) | 2.12 (2.90%) | −1.30 (−1.78%) |
| Very hot spot (99% of confidence level) | 10.82 (14.83%) | 9.42 (12.91%) | 10.35 (14.18%) | 7.95 (10.89%) | −2.87 (−3.93%) |

## 4. Discussion

### 4.1. Urbanization: An Assessment for Effective Urban Planning

This work assessed the seasonal (summer and winter) thermal state over the city landscape of Prayagraj city (India). The effects of land indices, namely, NDBI, EBBI, NDMI, NDVI, NDWI, and SAVI, on the thermal state were extensively examined to investigate how water bodies, forest land, wetland, and barren soils control intensification and/or cooling of LST over the landscape of the study area. The study area was delineated using eight-directional ring profiling of land indices, including LST to explore how, where, and what magnitude the LST changed either in increasing or in decreasing patterns due to different land indices. This can help policymakers and planners to conduct sustainable planning and enrich the carrying capacities of the landscape. As per the IPCC AR6 report of 2021, at the local to the global level, an extreme transformation of landscape has occurred, especially in the postindustrial era, which needs to be monitored and mitigated to control the rise in global mean temperature, leading to a decrease in the adverse consequences of climate change [5]. In connection to this issue, multiple cities (such as Taipei city of Taiwan [15], Phoenix city of the United States of America (USA) [16], Singapore [10], Dhaka city of Bangladesh [17], Kathmandu valley of Nepal [18], Nanjing city of China [19], Beijing city of China [20], Tokyo city of Japan [21], Tehran city of Iran [13], 70 selected cities of Europe [22], Hong Kong [23], Baltimore–DC metropolitan area of the USA [24], and Cairo city of Egypt [25]) have witnessed a similar amplifying pattern, which is very concerning and critical for our blue planet. UN-Habitat (2018), through SDG-11, categorically acknowledged the significance of spatial identification of land coverage and their possible effects on the safety, resilience, and sustainability of the city landscape, especially using greenery and open spaces [37].

Worldwide, to minimize the adverse effects of SUHI, including the rise in energy consumption, water scarcity, air pollution, and health problems, e.g., sunstroke, cardiac, and respiratory issues [68,69], populaces are converging on different mitigation strategies, such as the use of light paints and materials, planting of trees on streets, and cool and green roof creations [70,71]. Other strategies, such as designing the size, orientation, and shape of buildings, could improve the wind flow in the city landscape [72]. Moreover,

distinctive mitigation approaches, such as preserving wetlands, small to large water bodies, and greenery plantations on barren spaces, can significantly improve the local climate and ecosystem of the city landscape [38,73,74]. For this strategy and subsequent planning and creation, remote sensing and GIS-based information using a multitemporal spatial database can be applied to recognize the ground reality. Along with this, public awareness regarding the adoption and implementation of the above strategies is a prerequisite for their success in sustainable development and ecosystem restoration of the Prayagraj city landscape.

### 4.2. An Overview of Night-Time LST for SUHI Exploration

The spatiotemporal summer night-time LST maps are shown in Figure 19a, where summer time points, such as May 2008 and May 2018, were incorporated to present the night-time LST dynamics along with their whisker boxplot statistics. The mean night-time LST was severely intensified by 6.94 °C during the summer time periods of 2008–2018. At night-time, the central urban core area intensively experienced higher LST than its periphery.

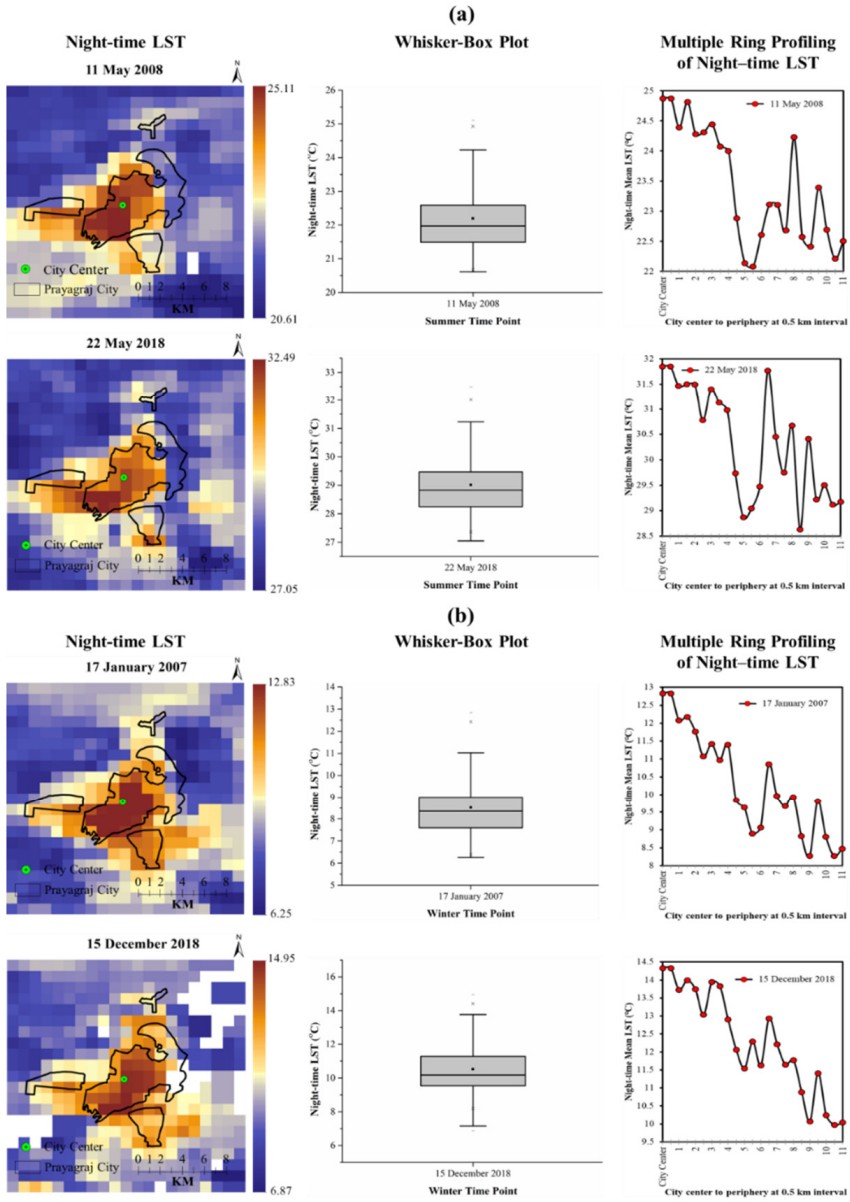

**Figure 19.** Maps of night-time LST dynamics with whisker boxplots for exploring SUHI at night-time: (**a**) summer and (**b**) winter.

The spatiotemporal winter night-time LST maps are shown in Figure 19b. Winter time points, such as January 2007 and December 2018, were incorporated to present the night-time LST dynamics and their whisker boxplot statistics. The mean night-time LST was also intensified by 1.69 °C during the winter periods from 2007 to 2018. At night-time, the central urban core area intensively experienced higher LST than its periphery. It was further detected that the SUHI phenomenon at night-time was severely intensified by 0–2.98 °C and 0–4.56 °C in summer and winter, respectively, according to multiple ring profiling. Other major cities over the Indo-Gangetic plain, such as Delhi [34], Lucknow [30], Patna [36], and Kolkata [35] have also been experiencing a strong SUHI phenomenon, whereby the city center has exhibited a higher LST than the periphery in the last few decades; a similar pattern was observed in our study area.

The above findings of night-time LST dynamics indicate a severe SUHI occurrence in both the summer and the winter seasons at night-time. These findings further strengthen the results of the daytime LST of Landsat imagery from 2008 to 2018. In the daytime, impervious surfaces absorb albedo extensively, and a long time is taken to radiate it back to the atmosphere due to its physical properties. As a result of increasing urbanization, this phenomenon has been further intensified. Both daytime and night-time LST experience a strong SUHI phenomenon; however, at night-time, SUHI becomes more vulnerable. As per this study, it is suggested to take immediate attention to reduce the SUHI severity using an effective mitigation strategy after urban planners and policymakers consider the aforementioned spatial thermal anomalies in the city.

## 5. Conclusions

This study explored the dynamics of seasonal (summer and winter) land indices [namely, NDBI, EBBI, NDMI, NDVI, NDWI, and SAVI] and LST dynamics using Landsat 5 (TM) and Landsat 8 (OLI/TIRS) imagery for Prayagraj city of India. The multitemporal spatial pattern of land indices and LST and their correlation dynamics with directional ring profiling over the city landscape were investigated, including the formation of SUHI and its dynamics over the city landscape. It was found that summer periodical LST magnitudes were highly intensified during S1–S4 by 0.32–2.45 °C (except for zone 11 km) at 0.5 km intervals from the city center to the periphery. Winter periodical LST magnitudes were also intensified by 0.02–1.06 °C (except for zones 9, 11, and 15–19 km). It was witnessed that the northeast and southwest directions had a high growth of LST distribution, while the northwest direction had a low growth of LST distribution in both seasons.

The results based on directional ring profiling of the effect of land indices on LST found that most of the vegetation/forest land available, at 1–3 km in the northwest and 5–6 km in the southeast direction, were depleted during the selected period. The impervious/built-up land expanded from the city center to 8 km in all directions during the study time, whereas bare soils and sand were primarily present in the northeast and the northwest (6–11 km). The presence of different land covers significantly controlled the LST distribution as forested area decreased the LST distribution whereas built-up area, bare soils, and sands increased the LST distribution.

Forest cover played a crucial role in declining the LST by 2.25–4.8 °C (except for water bodies), whereas bare soils and sand played a critical role in amplifying the LST by 1.9–5.6 °C. At the same time, built-up land amplified the LST by 1.8–3.9 °C (except for sand and bare soils). These results were further strengthened by the findings that LST vs. NDVI, LST vs. SAVI, LST vs. NDMI, and LST vs. NDWI had a positive correlation in the summer season. However, the LST vs. NDWI relationship had a very weak positive correlation due to the unavailability of water bodies of a significant size, which may have reduced the LST. In contrast, LST vs. NDBI and LST vs. EBBI had a strong positive correlation.

In the winter season, a positive correlation was observed for LST vs. NDVI, LST vs. SAVI, LST vs. NDMI, and LST vs. NDWI. However, the NDVI vs. LST, SAVI vs. LST, and NDWI vs. LST relationships showed a weak positive correlation to reducing mean LST. In contrast, EBBI vs. LST and NDBI vs. LST had a strong positive correlation. These

findings indicate that forest and water body coverage played a vital role in reducing the LST, whereas bare soils and sands played a substantial role in amplifying the LST in the Prayagraj city landscape.

The SUHI results confirmed that the urban center observed a higher LST than rural/suburban points in the range of 0.398–4.016 °C and 0.45–2.24 °C in the summer and winter, respectively. Furthermore, according to the directional ring profiling analysis, it was detected that the center of the city had a higher LST than the periphery up to 11 km, mainly in the northwest and the southeast (except for zones 6 and 8 km) directions by 0.1–4.1 °C and 0.1–4.5 °C in the summer and winter seasons, respectively.

Hotspot analysis (using Getis–Ord $G_i^*$ statistics) revealed that very cold spot, cold spot, and cool spot areal coverage declined over the study period in both the summer and the winter seasons. Hotspot analysis revealed the forested and Ganga River areas for very cold, cold, and cool spots, which were also observed in the directional ring profiling of land indices. This further strengthens our findings. It was further detected that the SUHI phenomenon at night-time was severely intensified by 0–2.98 °C and 0–4.56 °C in the summer and winter seasons. Therefore, to minimize the adverse effects of LST intensification and environmental sustainability of the local climate and ecosystem in the Prayagraj city landscape, the aforementioned delineated spatiotemporal seasonal thermal state and the hot and cold spot areas need to be prioritized by urban planners and policymakers for the design of suitable mitigation strategies.

**Supplementary Materials:** The following supporting information can be downloaded at: https://www.mdpi.com/article/10.3390/rs15010179/s1.

**Author Contributions:** Conceptualization, M.O.S. and R.D.G.; methodology, M.O.S.; software, M.O.S.; validation, M.O.S.; formal analysis, M.O.S.; investigation, M.O.S.; resources, M.O.S. and R.D.G.; data curation, M.O.S.; writing—original draft preparation, M.O.S.; writing—review and editing, M.O.S., R.D.G. and Y.M.; visualization, M.O.S. and R.D.G.; supervision, R.D.G.; project administration, M.O.S. and R.D.G.; funding acquisition, Y.M. All authors have read and agreed to the published version of the manuscript.

**Funding:** This research was partly supported by the Japan Society for the Promotion of Science (JSPS) grants of 21K01027 and 18H00763.

**Data Availability Statement:** Not applicable.

**Acknowledgments:** The authors are thankful to the USGS for providing the Landsat datasets for daytime seasonal variation and MODIS datasets in the night-time. The authors are also thankful to NASA Langley Research Center (LaRC) POWER Project for weather information. The first author is grateful to the UGC for providing a financial assistantship through a Maulana Azad National Fellowship for Minority Students (MANF) by the Ministry of Minority Affairs during his PhD research (Award Letter No. F1-17.1/2017-18/MANF-2017-18-WES-84175/(SA-III/Website)).

**Conflicts of Interest:** The authors declare no conflict of interest. The funders had no role in the design of the study, in the collection, analyses, or interpretation of data, in the writing of the manuscript, or in the decision to publish the results.

**Appendix A**

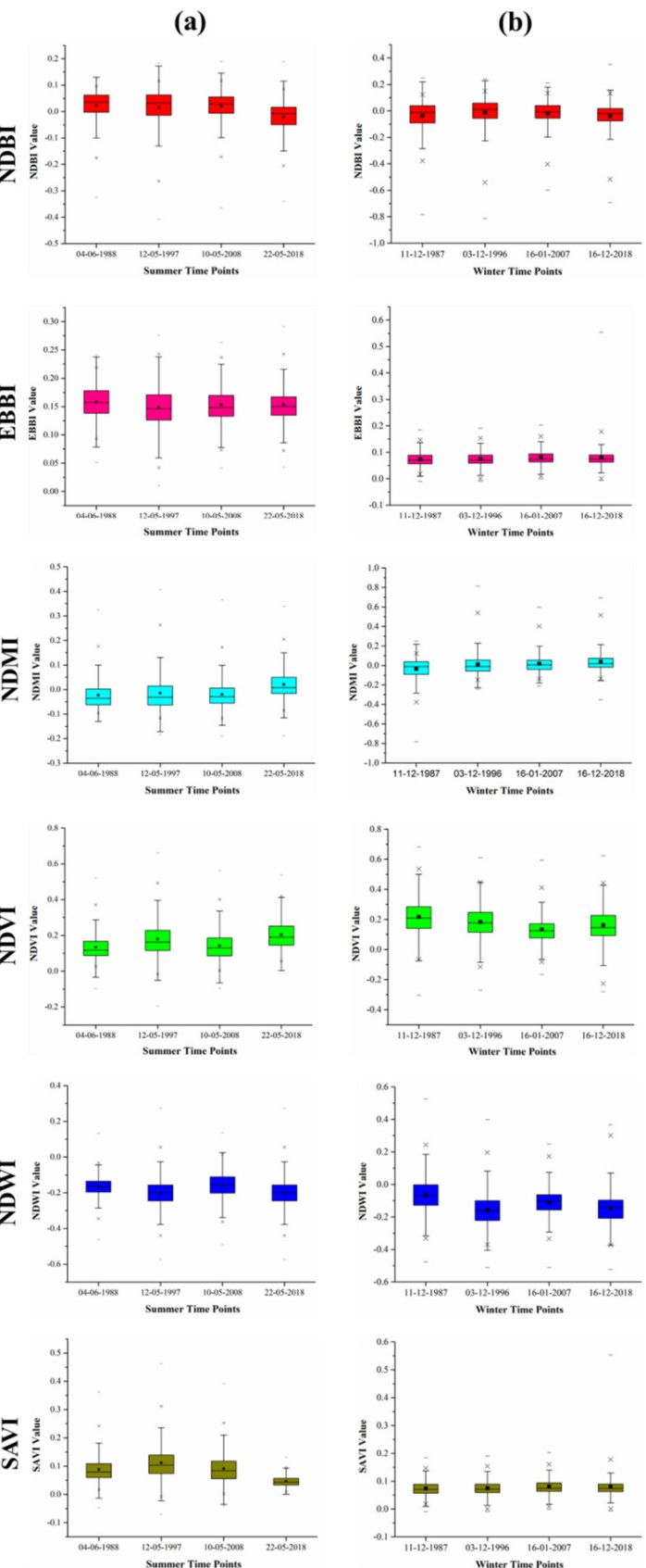

**Figure A1.** Seasonal whisker boxplots of the six land indices: (**a**) summer and (**b**) winter.

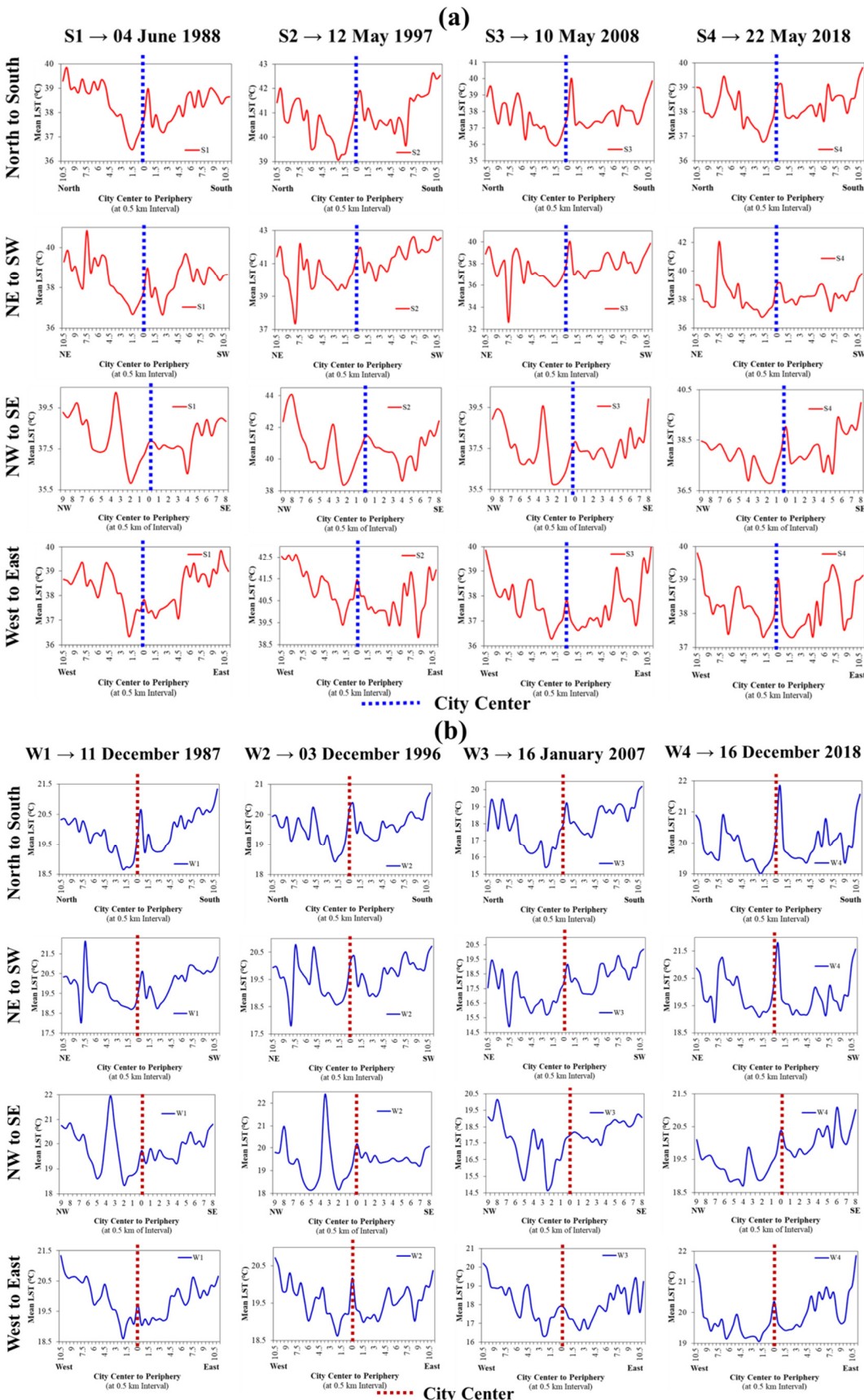

**Figure A2.** Seasonal LST profiling for SUHI formation in Prayagraj city (1987–2018): (**a**) summer and (**b**) winter.

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
