# Peer review of "Assessing Local Climate Change by Spatiotemporal Seasonal LST and Six Land Indices, and Their Interrelationships with SUHI and Hot–Spot Dynamics: A Case Study of Prayagraj City, India (1987–2018)"

_remotesensing, doi:10.3390/rs15010179_

Round 1

Reviewer 1 Report

The paper entitled ‘Assessing Local Climate Change by Spatiotemporal Seasonal LST and Six Land Indices, and their Interrelationships with SUHI and Hot–Spot Dynamics: A Case Study of Prayagraj City, India (1987–2018)’ deals with the seasonal pattern of LST and surface biophysical parameters in an ancient city. The study uses Landsat datasets for the analysis of LST and surface biophysical parameters. The study has been supported with very nice diagrams and figures. The methods used are relevant and study has been supported with a proper literature review. Results and discussions are also presented nicely. The article may be accepted after some minor corrections suggested below.

As the article deals with LST and SUHI of a city of Indo-Gangetic plains, I feel that authors should compare the results with the similar studies on other major cities of Indo-Gangetic plains like Delhi, Lucknow and Kolkata.

Secondly, The LST validation is an important step of UHI analysis, so, author should discuss about the validation in the method section.

Line 102, author says that the interrelationships between land indices and LST dynamics has been validated using Google Earth. I think this sentence should be corrected as LST validation is not possible using Google Earth.

Section 2.3.2.1, authors should discuss the reason why they have used MWA for LST retrieval. Some articles comparing the algorithms (such as https://doi.org/10.1007/s12517-020-06068-1 and https://doi.org/10.3390/s19225049) may be referred here.

There are so many figures in the articles. I think some figures may be shifted to the appendix.

Please check the following articles on SUHI and LST of some cities of Indo-Gangetic plains:

·        Land use/land cover change and its impact on surface urban heat island and urban thermal comfort in a metropolitan city. https://doi.org/10.1016/j.uclim.2021.101052

·        Impact of land use change and urbanization on urban heat island in Lucknow city, Central India. A remote sensing based estimate. https://doi.org/10.1016/j.scs.2017.02.018

·        Examining the expansion of Urban Heat Island effect in the Kolkata Metropolitan Area and its vicinity using multi-temporal MODIS satellite data. https://doi.org/10.1016/j.asr.2021.11.040

·        Surface Urban Heat Island (SUHI) Over Riverside Cities Along the Gangetic Plain of India. https://doi.org/10.1007/s00024-021-02701-6

Author Response

Dear Reviewer-1,

Please find our response in the attached file.

Reviewer 2 Report

The manuscript “Assessing Local Climate Change by Spatiotemporal Seasonal LST and Six Land Indices, and Their Interrelationships with SUHI and Hot-Spot Dynamics: A Case Study of Prayagraj City, India (1987-2018)” has been reviewed. The article investigates the interrelationship between LST and six land indices (NDBI, EBBI, NDMI, NDVI, NDWI and SAVI), as well as the dynamic analysis of SUHI and hot-Spot. In the analysis section, the selected impact factors are many and the workload is sufficient, but the study is not innovative enough, and I must therefore reject it. I have made some suggestions for changes that you can refer to.

(1) The article wants to study the climate change during 32 years from 1987-2018, but only 4 days of data were selected, which is not representative, and it is suggested to add time series analysis.

(2) Suggest a textual description of Figure 2.

(3) Chapter 3.2 mentions that “a declining trend was observed in the period of S1-S2.”, which is different from the trend shown in the subsequent Table 5, please check.

(4) There are many charts in each section of the article, which take up a lot of space, but some of them are not very useful in the article. For example, Figure 8 is mentioned only once in the text and is not analyzed, so it is suggested that such charts can be appropriately deleted.

(5) Please explain why the distribution of data points in the first three columns of the graph in Figure 15 is discontinuous, unlike the fourth column and Figure 14.

(6) In chapter 3, the article mentions water bodies/ forest land/ wetland/ barren soils and other feature types several times, and it is suggested that a map of the distribution of surface types in the study area be added to the article.

(7) Chapter 4.2 is proposed to be placed in chapter 3.

(8) Please add discussion to compare the LST spatio-temporal variation results and other research.

(9) Please validate the climate change (1987-2018) in your research through collecting the previous happened events in the study areas.

Author Response

Dear Reviewer-2,

Please find our response in the attached file.

Reviewer 3 Report

The paper "Assessing Local Climate Change by Spatiotemporal Seasonal LST and Six Land Indices, and Their Interrelationships with SUHI and Hot–Spot Dynamics: A Case Study of Prayagraj City, India (1987–2018)" is aproaching an interesting topic, and the authors did a lot of work.

In our opinion, if you want to analyse the Urban Heat island , then it is neccessary to show the surrounding of the area. When you have used Modis images, you showed us the island in a larger spatial context , but in the other cases it is actually imposible to understand and correlate the text with the images.

We strongly consider that you need to recreate the maps with the surrounding area, in order to be possible to visualize your results.

You can mantain the actual limit as an over lay, you can also remain with the same statittical result, but you need to show us the entire area, ant to explain why you are eliminating some areas.  If the results of image processing make this request difficult, you can overlay a mask on the problematic area.

Bassically the paper is good, and with the modification of all the maps from figure 3-4 and fig.9 to 13, and to make the modification in the text acording to the new result, the paper can be published. Thank you! Congratulation

Author Response

Dear Reviewer-3,

Please find our response in the attached file.

Round 2

Reviewer 2 Report

The author's response has addressed some of the changes I suggested. It is suggested that the authors put some of the figures in an appendix, which does not need to take up a lot of space.

Author Response

Dear Reviewer-3,

With due respect, please find the attached file for our response in response to to your comments and suggestions.

Thank you so much for your positive feedback.

Reviewer 3 Report

The paper was improved and can be published in present form, Congratulations!

Author Response

Dear Reviewer-3,

Thank you so much your valuable comments and suggestions. We are very thankful and grateful for your kind consideration to accept our work for publication in very reputed journal.